# Rethinking the Esterquats: Synthesis, Stability, Ecotoxicity and Applications of Esterquats Incorporating Analogs of Betaine or Choline as the Cation in Their Structure

**DOI:** 10.3390/ijms25115761

**Published:** 2024-05-25

**Authors:** Marcin Wysocki, Witold Stachowiak, Mikołaj Smolibowski, Adriana Olejniczak, Michał Niemczak, Julia L. Shamshina

**Affiliations:** 1Chair and Department of Inorganic and Analytical Chemistry, Poznan University of Medical Sciences, Rokietnicka 3, 60-806 Poznan, Poland; marcin.wysocki@student.ump.edu.pl; 2Faculty of Chemical Technology, Poznan University of Technology, Berdychowo 4, 60-965 Poznan, Poland; witold.stachowiak@doctorate.put.poznan.pl (W.S.); mikolaj.smolibowski@gmail.com (M.S.); adriana.olejniczak@doctorate.put.poznan.pl (A.O.); 3Fiber and Biopolymer Research Institute, Department of Plant and Soil Science, Texas Tech University, Lubbock, TX 79409, USA

**Keywords:** quaternary ammonium salts, cleavable compounds, esters, environmental safety, cationic surfactants, biological activity

## Abstract

Esterquats constitute a unique group of quaternary ammonium salts (QASs) that contain an ester bond in the structure of the cation. Despite the numerous advantages of this class of compounds, only two mini-reviews discuss the subject of esterquats: the first one (2007) briefly summarizes their types, synthesis, and structural elements required for a beneficial environmental profile and only briefly covers their applications whereas the second one only reviews the stability of selected betaine-type esterquats in aqueous solutions. The rationale for writing this review is to critically reevaluate the relevant literature and provide others with a “state-of-the-art” snapshot of choline-type esterquats and betaine-type esterquats. Hence, the first part of this survey thoroughly summarizes the most important scientific reports demonstrating effective synthesis routes leading to the formation of both types of esterquats. In the second section, the susceptibility of esterquats to hydrolysis is explained, and the influence of various factors, such as the pH, the degree of salinity, or the temperature of the solution, was subjected to thorough analysis that includes quantitative components. The next two sections refer to various aspects associated with the ecotoxicity of esterquats. Consequently, their biodegradation and toxic effects on microorganisms are extensively analyzed as crucial factors that can affect their commercialization. Then, the reported applications of esterquats are briefly discussed, including the functionalization of macromolecules, such as cotton fabric as well as their successful utilization on a commercial scale. The last section demonstrates the most essential conclusions and reported drawbacks that allow us to elucidate future recommendations regarding the development of these promising chemicals.

## 1. Introduction

Quaternary ammonium salts (QASs or “quats”) are commonly known as organic compounds that incorporate a positively charged nitrogen atom substituted with four alkyl substituents as a cation and any compatible counterion as the anion. Historically, the term “esterquat” was first used in 1931 in a patent application to define a specific quaternary ammonium salt that possessed an ester linkage within the the cation [1,2]. A few examples of the popular esterquats are shown in Figure 1, with ester linkages highlighted in different colors.

The ester bond within esterquats’ structures facilitates their hydrolysis and biodegradation, which allows chemists to perceive them as more environmentally friendly compounds compared to QASs bearing saturated or unsaturated alkyl chains on their nitrogen atoms, and containing only C–C and C–N bonds within the cation [3,4,5]. For instance, both the chemical and enzymatic hydrolysis of betaine esterquats yield glycine betaine and fatty alcohol, both products that can be regarded as environmentally safe and possibly even edible [2]. Consequently, esterquats are capable of demonstrating lower toxicity and persistence in the biosphere in comparison to classical QASs. However, this property varies widely depending on the type of esterquat and the number and positions of the appended functional groups; therefore, one should be aware of creating strict, one-sided generalizations devoid of objectivity [6]. Because this argument is one of the cautions in this field, the question ‘what is the real relationship between structure of esterquat and its ecological profile?’ will be thoroughly discussed further in this review.

Nowadays, esterquats are being considered as a unique class of compounds with highly beneficial properties that highlight their tremendous application potential. They are much more popular than a few decades earlier because of the societal shift to a circular economy, and efforts to save the planet. It was noted that due to potentially superior eco-toxicological profiles (particularly susceptibility to biodegradation and lower toxicity to aquatic live), esterquats are beginning to replace some QASs in both Europe and the United States [7]. Specifically, triethanolamine esterquat (TEAQ, Figure 1) has a primary energy demand (PED) of around 40 GJ/ton, whereas for most of the ”quats” this parameter is generally ~30–90% greater, 52 to 77 GJ/ton [8]. Despite the fact that esterquats are well-known, their properties can be easily designed to satisfy specific performance requirements, providing multiple promising areas for their future development that are unexplored so far. Various applications of esterquats, readily described and found in the framework of this review are summarized in Figure 2.

In 2015, the global market size of esterquats was estimated at USD 1.35 Billion and is forecasted to grow with a Compound Annual Growth Rate (CAGR) of 11%, during the period from 2016 to 2024 [28]. This trend also results from the current, more stringent requirements set by customers, who much more frequently select less noxious products for humans and the environment. Currently, the most popular representatives of esterquats (by volume of production) are based on triethanol amine (TEAQ), diethanol amine (DEEDMAC), or *N*,*N*,-dimethyl-3-aminopropane-1,2-diol (HEQ) (structures are shown in Figure 1) and are utilized mainly as surfactants and fabric softeners [29,30,31,32,33]. Recently, Dow Global Technologies LLC, Dow Silicones Corp, and Rohm & Haas have patented an innovative fabric care product containing esterquat TEAQ and quaternary ammonium functionalized dextran polymer facilitating its deposition on the fabric [34]. Early uses of esterquats also include textile auxiliaries and dye leveling agents [3], because they enable a more effective dyeing processing operation, stabilizing the dyeing bath to achieve uniform fabric shade.

Another interesting application refers to so-called flocculants, compounds that promote the aggregation of suspended particles in fluids. The growing need for flocculants in the mining industry and water treatment is currently driving the revenue growth of the flocculants market, which relies on a specific group of esterquats as well [35].

Analysis of the patent databases revealed another significant use of esterquat-type compounds referring to the enormous and rapidly developing market of personal care. Among various household products, esterquats of diverse compositions are used as components of creams and lotions, shampoos, conditioners, and skin care products [1,36,37]. A key aspect of novelty in these inventions is attributed to improved safety for customers and/or more benign environmental profiles compared to that of classic QASs, which are intended to be replaced. The increased interest in esterquats from the largest chemical companies, such as BASF, Henkel, Procter Gamble or Evonik serves as an indicator of their great usability and sufficient safety, which would potentially translate to their widespread commercialization and scale-up production [37,38,39,40,41].

Interestingly, the pH-labile linkages in esterquats allow their considerations as potentially useful agents for special applications in the biotech industry, such as biocides (including both antiseptics and disinfectants), drug delivery vehicles, and gene therapy agents [2]. Thus, some of the systems based on esterquats were found to be suitable for systemic delivery and for intra-tumoral injection of chemotherapy into solid chemotherapy-resistant tumors. It is hypothesized that esterquats are able to form appropriate compact spherical nanoparticles when combined with siRNA and are able to deliver the siRNA to cancer cells where it induces gene silencing [42]. Recently, cationic solid lipid nanoparticles with cholesterol-mediated esterquat DEEDMAC (Figure 1) have proven to be effective in the controlled release of protease inhibitor Saquinavir (Invirase^®^) without inducing significant endothelial toxicity, which reveals their surprisingly good biocompatibility with human brain-microvascular endothelial cells [43].

Due to the fact that the use of conventional pesticides causes many negative effects, including environmental contamination (soil, water bodies, atmosphere, living organisms at various trophic levels, etc.) and pest resistance, various initiatives have been established to minimize these events, such as the OECD Programme on Pesticides and Sustainable Pest Management [44]. Recently reported esterquats derived from commercialized herbicides (e.g., 2,4-dichlorophenoxyacetic acid (2,4-D), 3,6-dichloro-2-methoxybenzoic acid (dicamba), 2-methyl-4-chlorophenoxyacetic acid (MCPA), or iodosulfuron-methyl (IS-M)) have proven to be an attractive replacement to off-the-shelf formulations, in terms of efficacy, safety to operator, as well as improved environmental impact [11,12,14]. The importance of this segment is justified by statistical data, according to which the global herbicides market size amounted to USD 36.93 Billion in 2021, and was anticipated to grow with a CAGR of 6% during subsequent years [45].

Esterquats also exhibit a tremendous potential to be applied in other important fields, such as electrochemistry where they are used in various applications, from corrosion inhibitors to energy storage media in batteries and electrochemical capacitors [27,46]. It should also be emphasized that these unique compounds are being increasingly utilized as a source of cations for the synthesis of ionic liquids (ILs) or hydrogen bond acceptors (HBA) for the synthesis of Deep Eutectic Solvents (DESs), respectively. These types of systems could be appropriately designed to exhibit a lower impact on ecosystems in comparison to marketed chemicals or demonstrate superior physicochemical properties ensuring their successful application [9,12,47,48,49,50].

Despite their versatility and numerous advantages, a literature survey reveals only two mini-reviews discussing the subject of esterquats. The first one appeared in 2007 and briefly summarized published reports regarding esterquats’ types, synthesis, and structural elements required for a good environmental profile and briefly covered applications [1]. Obviously, more than 15 years is a sufficient period of time for certain data to be reevaluated. In the second review article, the stability of selected betaine-type esterquats in aqueous solutions was reviewed, whereas other aspects were not covered. Additionally, one article comparing the toxicity of various surfactants, involving esterquats was also published [51]. Herein, we provide a current and thorough set of data that will help design and efficiently synthesize novel esterquat-type structures. Additionally, the susceptibility to hydrolysis of both choline-type and betaine-type esterquats is thoroughly reviewed, as it is a crucial factor in terms of the majority of applications and their safety. Further sections are devoted to the analysis of the biodegradability and toxicity of esterquats, which allow chemists to design compounds that are both safer in use and do not pose a threat to the environment. In the last part of this review, the various currently known applications of esterquats as well as perspectives and future research directions regarding their development are discussed.

## 2. Types of Esterquats

In regards to their chemical structure, two main types of esterquats can be distinguished based on the arrangement of chemical bonds in the ester group: choline-type esterquats (esters of carboxylic acids and quaternary aminoalcohols) and betaine-type esterquats (esters of quaternary amino acids and alcohols) [3,6], Figure 3.

### 2.1. Synthesis of Choline-Type Esterquats

In the last decade, biobased esterquats were extensively explored in pursuit of nontoxic, naturally occurring and biodegradable cations, such as choline or its acylated derivatives (e.g., acetylcholine) [7,13,52]. Choline is widely known as a source of methyl groups required for various stages of animal and plant metabolism; this ion is an essential nutrient that is naturally present in food and available as a dietary supplement [53,54]. Generally, living cells require choline to preserve their structural integrity and to produce acetylcholine, an important neurotransmitter for various functions of the brain and nervous system, such as memory, emotion control, and muscle control [53,54]. The excellent eco-toxicological profile of commercialized choline-based cations (such as TEAQ and DEEDMAC) along with wide availability and relatively low price are the main factors contributing to their frequent selection for the synthesis of various derivatives, including a large group of choline analogs comprising an ester bond, esterquats [55,56,57]. In addition, there are multiple ILs and DESs [58,59,60] derived from esterquats. Generally, some ionic esterquats with melting points lower than 100 °C can be classified as ILs [14], whereas others can be used as hydrogen bond acceptors in DESs [15]. ILs and DESs share a mutual advantage, specifically designability, which can be defined as the possibility to tune their properties to the needs of an expected application. This can be conducted by selecting the most appropriate ions in the case of ILs or combining the proper HBA and HBD in the case of DES systems [15].

Esterquats are characterized by a diverse chemical structure, with the ester group (-O-C(O)-) located between the choline residue and a linear alkyl chain [3,61,62]. In some other instances, the choline cation and biologically active carboxylic acid (such as herbicide) can be bound together via the formation of an ester group that serves as a linker (Figure 3) [9,47,63]. Detailed data on the synthesis of choline-type esterquats, including the list of reagents, brief procedures for product isolation and purification as well as product yields are provided in detail in the Appendix A, in the attempt to provide sufficient information to readers who are enthusiasts of organic synthesis. Below, we summarized the general methods of their synthesis (Table 1).

### 2.2. Potential of Choline-Based Esterquats Synthesized from Biologically Active Compounds

#### 2.2.1. Herbicidal Formulations

Multiple reports have revealed the successful synthesis of esterquats comprising biologically active moieties via creating an ionic bond between esterquats cations and biologically active anions [9,47]. The derivatization of biologically active compounds toward esterquats can be considered a potentially promising strategy providing access to novel, enhanced forms of already-known active ingredients.

Popular herbicides, such as 2,4-D, dicamba, and MCPA serve as excellent examples, wherein it was possible to obtain the biologically active salt form with simultaneous elimination of herbicides’ greatest weakness, volatility (please see data in Table 2) [12]. Additionally, the derivatization of known herbicides toward esterquats potentially allows for the modification of their affinity for water and eliminates the threat of contamination of aquatic and soil ecosystems with highly water-soluble herbicides. Even more, the selection of herbicidally active counterions permits the formation of “dual functional” herbicides via this approach. These “dual functional” herbicides broaden the spectrum of herbicidal activity, enhance efficacy and thus diminish the possibility of the emergence of herbicide-resistant weeds [9].

As was shown in Figure 4, there are several ways to insert a herbicide into an esterquat structure. The formation of salts or IL from the cholinium cation and herbicidal anion introduces the herbicide via ionic bonding (Strategy A, Figure 4). In this strategy, the esterquat anion is replaced with a herbicidal anion via an anion exchange reaction. Such an approach enables “tuning” of the physicochemical properties of the formed products while introducing additional biological activity brought about by herbicides [85,86]. The reason to use anion exchange over the acid-base reaction is to avoid issues associated with the hydrolysis of an ester bond and decomposition of the cation (The issue of hydrolysis of an ester bond in esterquats is thoroughly described in Section 3).

At the same time, adjustment of unfavorable properties of the herbicide (e.g., volatility) can be conducted through covalent modifications of the esterquat structure, similar to the prodrug approach, with a formation of easily hydrolyzable ester bond (Strategy B, Figure 4). Finally, a combination of salt formation and a “prodrug” strategy can be used (Strategy C, Figure 4). Table 2 illustrates the examples of prepared herbicides.

Currently, the synthesis of these compounds is well described. There are also data available on their herbicidal activity, biodegradability, and toxicity toward soil microorganisms (described in Section 4). However, further studies about their environmental impact, such as aquatic toxicity tests, should be made.

#### 2.2.2. Other Biologically Active Systems

Esterquats provide new insights into drug delivery systems. Thus, the state-of-the-art reveals an interesting group of dual-functional esterquats exhibiting both enhanced tissue permeation and analgesic activity and derived from carbonate and lidocaine moieties [84]. For example, the lidocaine-derived choline was linked with carbonate and fatty alcohol to obtain asymmetric gemini surfactant (Figure 5).

The carbonate moiety was chosen over “classic” ester because of its amphiphilicity which allows it to serve as a second hydrophilic head. As a result, the enhanced hydrophilicity of the compounds permitted improved tissue permeation. Surprisingly, the compounds showed spontaneous self-assembly into specific microstructures, known as Janus particles [84]. This kind of agglomerate was known before to be prepared mainly from macromolecules or multi-components systems [87]. The report demonstrated that smaller organic molecules have the ability to self-assemble into such systems subject to meeting certain requirements such as the existence of an aromatic group (present in lidocaine moiety) and an amphiphilic carbonate group. The lengths of alkyl chains (the spacer and the hydrophobic tail) in the molecule were not as important. Additionally, it was established that the amphiphilic carbonate groups are capable of increasing the tissue permeability of the whole particles. Such systems have the potential to be utilized in many fields. Because of lidocaine core and increased hydrophilicity, they exhibit unique interfacial and biological activity and can be utilized in gene transition, drug delivery systems, etc. (Figure 6, left, structures A and B) [84].

Systems aimed to deliver targeted drugs into cancer cells based particularly on choline analogs were successfully prepared and tested as well. One of the papers described the use of MDEA esterquat (Figure 6, middle, structure C, ester of oleic acid and diisopropanoldimethylammonium methyl sulfate) as a novel nanoparticle delivery system for cytostatic drugs and small RNA fragments (abbreviated as siRNA, small interfering RNA) [42]. Specifically, upon mixing with siRNA, MDEA formed spherical compact nanoparticles able to deliver the siRNA to cancer cells for induction of gene silencing. These systems were especially effective when the esterquat was combined with 1,2-dioleoyl-sn-glycero-3-phosphoethanolamine (DOPE) phospholipid, increasing transfection efficiency (i.e., the percentage of cells transfected from cells nontransfected), to 93%, enhancing the efficiency of the therapy [42].

One more study examined the anti-oxidant efficacy of sinapine-derived choline esterquats prepared from naturally occurring antioxidant sinapinic acid. Sinapine and its analogs are known to exhibit simultaneous antioxidant and antimicrobial activities, which was noted as potentially useful in the context of utilization in food preservation [57]. While the authors did not point out the advantages of the esterquat strategy, it can be presumed that the transformation of sinapine into esterquats likely enhanced the aqueous solubility of compounds and hence affected the release rate of active (Figure 6, right, structure D). Interestingly, it was the type of substituent that influenced the antioxidant activity of the prepared derivatives. Thus, in the case of hydrogen substituents (X = H, Y = H), no antioxidant activity was observed. The substitution of one or two hydrogen atoms with a methoxy group resulted in the enhancement of the antioxidant properties (X = OCH_3_, Y = H, EC_50_ = 36.7 nmol; X = OCH_3_, Y = OCH_3_, EC_50_ = 18.1 nmol), while the compound with the polyphenolic ring was characterized by the highest activity (X = OH, Y = H, EC_50_ = 6.3 nmol). The summary of biological activity for the compounds presented in this section is covered in Table 3.

The studies presented above unambiguously demonstrate the versatility of esterquats in the design and synthesis of various types of multifunctional esters that can be successfully applied in the majority of industries. Practically, there is no limitation regarding the type of biological activity that can be introduced into esterquat-based ionic pairs. Therefore, we presume that the number of similar literature reports describing this group of multifunctional compounds will be the subject of extensive studies by various research groups from both academia and industry. While the lack of toxicity data for these compounds limits the generalizability of the conclusions, it also presents a foundation for future research.

### 2.3. Synthesis of Betaine-Type Esterquats

In recent years, scientists have begun to shift their attention from choline to betaine-type esterquats, due to the development of efficient routes of their synthesis from cost-effective reactants. Recently, it has been shown that these esterquats can be synthesized directly from inexpensive, naturally derived glycine betaine or carnitine via simple *O*-alkylation under mild conditions [11,14,15,88]. Such an approach allows us to avoid not only the use of expensive but also toxic reagents that were required in former synthetic paths of this unique group of compounds. The simplest betaine, *N*,*N*,*N*-trimethylglycine, commonly known as glycine betaine, is omnipresent in living animals and plants, mainly due to its osmotic properties that prevent cell dehydration [89,90]. Both choline and glycine betaine are able to act as HBAs in the process of formation of natural deep eutectic solvents (NADESs), exhibiting tremendous potential for applications including those of an industrial scale [91,92,93]. Interestingly, betaine has been repeatedly mentioned in literature as a powerful hydrogen bond acceptor in DESs, but known reports describe the use of betaine in zwitterionic form, while its esters have not yet been explored for the purpose of DES formation. Choline-based DES and choline-type esterquats, where the anion contribution as HBA is crucial, are well-known. Theoretically, esterquats in the form of betaine esters, where the counterion is a chloride or bromide anion, may act analogously to choline chloride and be combined with a hydrogen-bond donor (HBD) (e.g., urea, amino acid, polyol, carboxylic acid, or sugar). It is possible that some of the formulated mixtures may turn out to be DESs, but such studies have not yet been conducted. Thus, the use of betaine esters as substitutes for choline chloride in the synthesis of DESs may constitute an interesting research direction in the coming years.

In humans, glycine betaine acts as a methyl group donor in the synthesis of methionine from homocysteine [94]. Glycine betaine is also utilized in crop production because it accumulates in many plant species under stress, which means that this compound can serve as an excellent platform for the synthesis of various promising agrochemicals [95]. It should also be emphasized that this poorly developed byproduct of the sugar industry, which constitutes up to 8% of sugar beet molasses [96], is readily biodegradable and practically nontoxic to living organisms (acute oral LD_50_ for rats ∼11,000 mg per kg allows classification of betaine into the 5th toxicity category according to Globally Harmonized System of Classification and Labelling of Chemicals (GHS)) [97]. Continuous discoveries of facile and environmentally friendly ways to access esterified derivatives of betaine are currently a driving force for a growing interest in this type of esterquats, which also leads directly to an increase in the reported spectrum of their innovative applications. In Table 4 we provide a summary of the general methods of synthesis of betaine-based esterquats (detailed data regarding the synthesis are available in Appendix A).

### 2.4. Potential of Betaine-Based Esterquats Synthesized from Biologically Active Compounds

Analogously to choline-derived esterquats, multifunctional, biologically active esterified forms of betaine are obtained via the anion exchange reaction routes [11,14,86]. Theoretically, hydrolyzable ester bonds in betaine derivatives can be utilized in drug delivery systems, wherein esterquats represent a platform for a controlled release of actives. However, thus far a literature analysis has not revealed any report on this subject.

Esterquats with a betaine-type cation (Table 5) have been successfully used for the synthesis of a new generation of agrochemicals. As a result, popular herbicides (e.g., iodosulfuron methyl and dicamba) have been successfully combined with alkyl betainate cations (Figure 7), where they constitute a biologically active anion. The cation in these ion pairs can be an excellent tool for modifying properties of parent herbicides, such as aqueous solubility or surface activity while exhibiting a low impact on the environment (the hydrolyzable ester bond facilitates the decomposition and degradation of such quaternary ammonium cations in the environment) [11]. Interestingly, in the case of some alkyl betainate cations (particularly wherein the alkyl chain contains ten carbon atoms), the cation itself cannot be considered as an inert affecting only the physicochemical properties, but as a phytotoxic substance, acting synergistically with the herbicidal anion [14]. Bearing in mind the low price and availability of glycine betaine which is a by-product in the sugar manufacturing process, the interest in this betaine as a starting material for novel, multifunctional and environment-friendly esterquats will likely intensify in the coming years.

## 3. Susceptibility to Hydrolysis of Choline-Type and Betaine-Type Esterquats

Two types of esterquats (choline-type and betaine-type), exhibit different physicochemical properties [3,6]. As demonstrated in Figure 8, the presence of an ester bond within esterquats structure makes them susceptible to hydrolysis, and these compounds are even more reactive than conventional esters [56,62,76,77]. The quaternary positively charged nitrogen atom in betaines pulls electron density from the neighboring atoms, resulting in carbonyl carbon becoming more electrophilic.

As a result, the hydrolysis of betaine esters is strongly pH-dependent. The higher electrophilicity of carbonyl carbon atoms provides betaine esterquats with good stability under acidic conditions but makes them more susceptible to hydroxyl attack [3,6,9,16,116]. Hence, under alkaline conditions, the rate of hydrolysis is higher than that of esters, in general; moreover, they hydrolyze at a significant rate by the base-catalyzed mechanism even at neutral pH. In an acidic environment, betaine esters are more stable than conventional esters [5,9,57,80,81].

The rate of hydrolysis is also dependent on the esterquat structure as well as on intermolecular interactions between ester and quaternary ammonium groups. In the case of choline-type esterquats, the distance between these groups is longer than for structurally similar derivatives of betaine. Moreover, in choline-type esterquats, nitrogen atoms are additionally separated from carbonyl carbon by oxygen. Therefore, the specific repulsions in betaine esters between the positively charged nitrogen and carbonyl carbon lead to the destabilization of the ground state, and in effect, such compounds tend to be relatively more stable in acidic conditions but hydrolyze more rapidly when the conditions are basic [5,16,57,78,80].

Gemini-type esterquats tend to be more susceptible to hydrolysis than their monocationic analogs [3,6,100]. It should also be noted that the orthoesters demonstrate the opposite tendency in terms of their susceptibility to hydrolysis, due to the lack of a carbonyl group in the cation, hence an increase in the pH of the solution results in the respective increase in stability [77].

Another important factor affecting the rate of hydrolysis of esterquats is the temperature of the solution. Obviously, a rise in temperature naturally increases the reaction rate, although for esterquats its impact remains relatively low in a wide spectrum of temperatures [122]. However, the rate of hydrolysis is also negatively influenced by the ionic strength of the solution. The addition of salts such as sodium chloride may significantly slow the process and make esterquats more stable to hydrolysis [16].

It was also established that hydrolysis of the esterquats in a basic environment often tends to accelerate the values of critical micelle concentration (CMC), which can be attributed to micellar catalysis. This phenomenon is explained by the increase in the concentration of hydroxyl ions near the micelles, caused by the occurrence of electrostatic forces between those ions and positive charges on the micelles’ surfaces [3,6,100]. Furthermore, it should be taken into consideration that the products of such hydrolysis might exhibit surface active properties (such as long-chain alcohols). This would result in a reduction in CMC and a simultaneous increase in the hydrolysis rate. On the other hand, the incorporation of the products of hydrolysis into those micelles slows hydrolysis due to the limited attraction of hydroxyl anions associated with the dilution of charges on the micelle surface. This charge dilution results in a lesser increase in pH, therefore, decreasing the rate of hydrolysis. However, this effect was found to be negligible compared to the reduction of CMC [3,6,100]. Another interesting factor affecting the rate of hydrolysis is the occurrence of side-chain groups between quaternary ammonium and ester groups. They can substantially decrease the rate of hydrolysis of such esterquats due to their hydrophobic nature [6].

The conditions, either aerobic or anaerobic (such as methanogenic conditions [81]) may possess a great impact on the products generated subsequently after the hydrolysis. The aerobic degradation of the hydrolyzed species can occur in various pathways, depending on the microbial strain and the enzymes involved in the process. Free choline released in the hydrolysis reaction is converted into glycine betaine, and then the process relies strictly on the degradation of betaine-related species. Unfortunately, during the process some toxic compounds such as formaldehyde may be produced, especially during the demethylation of dimethylglycine and then sarcosine to glycine. The leaving methyl group is often converted into formaldehyde by dimethylglycine dehydrogenase and sarcosine oxidase, respectively [123,124,125]. Another pathway that occurs mainly in the marine environment is based on the deamination of choline, betaine and related species into trimethylamine (TMA) via choline-TMA lyase and glycine betaine reductase, respectively. The formed TMA is then oxidized by TMA oxidase to trimethylamine oxide (TMAO), although the process is reversible due to the presence of TMAO reductase. On the other hand, TMAO can be further demethylated into dimethylamine (DMA) and then methylamine (MMA), also releasing formaldehyde, similarly to the betaine pathway. Then, MMA can be oxidized in several steps into another formaldehyde molecule that is accompanied by ammonia. Whether it is the main source of methylated amines in the marine environment has not been determined yet and still needs to be investigated [125,126].

Furthermore, many reports have revealed that esterquats (such as gemini-type cations containing both a carbonate linkage and a dodecyl group) can be effectively hydrolyzed with the use of various enzymes, such as lipase [5,81]. This means that under conditions where these compounds are known to be stable, it is still possible to influence their decomposition rate by using biotechnological tools.

The hydrolysis reaction depends on the initial esterquat structure. We have identified the core structures in Figure 9. The hydrolysis reaction generally proceeds under various conditions; the hydrolysis rate of esterquats and conditions employed are listed in Table 6 (the detailed structures are given in Appendix A).

## 4. Biological Tests

Esterquats, as a novel generation of active ingredients, exhibit significant potential for commercialization as a direct replacement of currently used compounds. Therefore, they can eventually end up in the environment, which includes mainly soil and watercourses. Hence, a comprehensive environmental risk assessment should be conducted to evaluate their toxicity toward living organisms as well as biodegradation in various conditions [1].

### 4.1. Biodegradation of Esterquats

To study the fate of esterquats in the environment, various biodegradation studies have been conducted. The collected data from the available reports are summarized in Table 7, and the detailed structures are given in Appendix A. It should be noted that biodegradation tests have not been performed for all esterquats described in the literature. This fact reveals a serious gap in crucial data that directly hinders their potential commercialization. Among the presented studies, standard protocols supported by The Organization for Economic Co-operation and Development (OECD) have been utilized, in particular OECD 310 (“Headspace Test”, conducted by evaluation of carbon dioxide produced by microbial action) and various variations of OECD 301 tests such as OECD 301 A (“DOC Die-away test”, conducted by measuring the change in Dissolved Organic Carbon (DOC)), OECD B (conducted by measuring the production of carbon dioxide (CO_2_) during its degradation), OECD 301 C (conducted by respirometry through measuring oxygen consumption), OECD 301 D (“closed bottle test”, conducted by respirometry through measuring amount of dissolved oxygen), OECD 301 E (conducted by evaluation of dissolved organic carbon) and OECD 301 F (conducted by manometric respirometry through measuring oxygen consumption) [3,12,19,61,69,81,82,111]. In these tests, the biodegradation percentage is obtained as the quotient of biological oxygen demand (BOD) and theoretical oxygen demand (ThOD) required to oxidize a compound to its final oxidation products [127].

In other cases, the OECD 310 protocol, also called the Ready Biodegradability Test, or carbon dioxide method, was used. The test of the level of biodegradation is based on the analysis of CO_2_ evolution resulting from the ultimate aerobic biodegradation of the test substance [128]. Interestingly, a direct comparison of the biodegradation test via OECD 301 C with other protocols proposed by OECD (OECD 301 A–F and OECD 310) was conducted in 2013 by Kayashima et al. [127,128,129] and revealed that substances evaluated as “not readily biodegradable” according to guideline 301 C might be “readily biodegradable” when tested via OECD 301 F and OECD 310 methods. Moreover, all selected tests applied to evaluate the biodegradability of the synthesized esterquats were based on anaerobic degradation with the use of activated sludge isolated from water plants, while potential biodegradation of herbicides in the environment will require significantly different microbiota.

For instance, as it was noted in a recent publication, research focused on microbial biodegradation of QASs in water and soil revealed substantial differences depending on the applied environment. The benzalkonium cation was readily mineralized in water, whereas the herbicidal 4-chloro-2-methylphenoxyacetate (MCPA) anion was able to fully degrade only in the soil environment [130]. These results indicated that esterquats comprising herbicidal anions should be tested by using microbiota isolated from different niches, where herbicides might occur due to applied agricultural practices.

As presented in Table 7 and Figure 10, the highest biodegradability level, up to 85–95% after 28 days, was demonstrated by monocationic esterquats with chloride and bromide anions (Table 7, Entries 26, 28 and 41) On the other hand, the lowest biodegradability (20%) occurred in the case of gemini surfactants composed of one oxycarbonyl group in the linker structure between two nitrogen groups (Table 7, Entry 29) and functionalized herbicides (2–12%, Table 7, Entries 13–19).

It was also shown that combining herbicidal anions with esterquat cations may lead to the formation of compounds with enhanced biodegradability compared to MCPA-free acid. In this study, the greatest biodegradability level was demonstrated by salt with di(tallowoyloxyethyl)dimethylammonium cation and MCPA anion whereas 2-[methacryloyloxy)ethyl]trimethylammonium salt with the same anion was significantly less biodegradable (Table 7, Entry 27: 63% vs. Entry 25: 29%) [12,52,131]. Obviously, such a scarce number of available results clearly demonstrates that there is still a substantial lack of data considering the biodegradation of esterquats incorporating herbicidal anions in their structures. Nonetheless, in the case of such structurally complex ionic pairs, it is recommended to analyze the decomposition of both ions separately rather than providing the sum of the biodegradation rate for the whole compound. Such action allows us to reveal the true environmental impact of each of the biologically active ions that were combined to form an esterquat molecule.

Interestingly, the analysis of the reported biodegradation studies revealed that the biodegradability of esterquats gradually decreases with an increase in the number of methylene groups between the oxycarbonyl group and the quaternary ammonium group, even when these surfactants possessed the same number of carbon atoms in the lipophilic part. The compounds with different types of cations are separated into **A**, **B**, **C**, **D**, **F**, **G**, **H**, **I**, **J**, **K**, **L**, **M**, and **N** classes based on their structural features (Figure 11).

Gemini-type compounds (**A**) with two methylene groups between the oxycarbonyl group and the ammonium group at the linker moiety presented higher biodegradation compared to the analog with four methylene groups (Table 7, Entry 5: 58% vs. Entry 11: 50%) [61]. Also, for most types of polycationic esterquats, biodegradation increased with an increase in the alkyl chain length connected to the nitrogen. Thus, for the gemini-type compound **G** elongation of two alkyl chains from decyl to dodecyl increases biodegradability from 60% to 70% (Table 7, Entries 30 and 31) [81]. Similar results were demonstrated for other tested tricationic esterquats (**N**) (Table 7, Entry 46: 41%; Entry 47: 42%; Entry 48: 52%) [111]. However, after reaching fourteen carbon atoms within the alkyl chain in esterquat **G**, the biodegradation was found to slightly decrease (Table 7, Entry 32: 60%) [81].

It should be noted that conventional gemini-type cations derived from hydrolytically cleavable moieties, such as ester or amide bonds, show almost no signs of biodegradation (pentane-1,5-bis(*N*,*N*-dimethyl-*N*-decylammonium) diiodide: 10%; pentane-1,5-bis(*N*,*N*-dimethyl-*N*-dodecylammonium) diiodide: 5%; pentane-1,5-bis(*N*,*N*-dimethyl-*N*-tetradecylammonium) diiodide: <5%) [81]. However, data for their analogs comprising at least one ester bond (**G** and **H**) revealed an affinity of such compounds to biodegradation in the OECD 301 C test that was several times greater (Table 7, Entry 33: 30%; Entry 34: 45%; Entry 35: 20% Entry 37: 25%) than that for conventional gemini-type cations [61,81].

Generally, it is an extremely challenging task to compare biodegradability results for esterquats containing cations characterized by such diverse chemical structures. Nonetheless, a thorough analysis of available studies shows that the alkyl chain length in the cation has a significant influence on their biodegradation independently of the applied testing method. Generally, alkyl chain elongation facilitates the affinity of compounds for microbial decomposition. It has also been established that increasing the number of methylene groups between the ester bond and the nitrogen atom causes a decrease in the biodegradability rate. Standard procedures developed by the OECD are being applied; however, their diversity makes it impossible to compare them directly. Moreover, due to the selection of different microorganisms, the results of the biodegradation test for herbicidally active esterquats might be unreliable taking into consideration their potential application. Therefore, it is crucial to develop novel standard procedures for evaluating the biodegradability of esterquats, particularly for substances with potential usage as herbicides.

### 4.2. Toxicity of Esterquats

Toxicity testing is indispensable for the overall sustainability characterization of chemicals as it ensures the responsible development and use of substances, safeguarding human health, and preventing environmental harm. That is why before the introduction of any chemical into the ecosystem, thorough toxicity studies should be performed in order to assess its potential impact, risk and possible interactions with the environment. Unfortunately, toxicity tests are rarely applied for the majority of newly synthesized esterquats. The situation is further complicated by the fact that there are no unified standard procedures for the purpose and only incidental studies have been performed. In the literature, we can find scarce data regarding specified, more sophisticated toxicity tests, such as the hemocyte-based bioassay. In this context, choline esters with middle-length straight alkyl chains and lactate or levulinate anions have shown negligible toxicity, but the lack of data for a broader group of esterquats makes it impossible to draw deeper conclusions [132]. Tests such as minimum inhibitory concentration (MIC [18,81,82]) and minimum bactericidal concentration (MBC [47,48]) are the most common parameters to evaluate the interactions of esterquats with microorganisms that are present in the environment. In some antimicrobial studies, the inhibition zone diameter (IZD [20,67]) was determined and half maximal effective concentration (EC_50_ [47,57]) tests were also conducted; however, such data are scarce. As presented in Table 8 (detailed structures are given in Appendix A), the toxicity testing of esterquats has been performed on common bacteria and fungi. There is still a lack of information regarding the toxicity of esterquats for other organisms, such as mammals, fish, shellfish, crustaceans, or algae. However, available data [12,21,111,133] indicate that they generally follow the same trends as conventional QASs [134]. Due to the different conditions of the performed tests and the manner of presenting the obtained toxicity data, it is difficult to provide indisputable conclusions.

#### 4.2.1. Antimicrobial Activity of Esterquats

The analysis of structurally different esterquats (The compounds with different types of cations are separated into **A**, **B**, **C**, **D**, **F**, **G**, **H**, **I**, **J**, **K**, **L**, **M**, **N**, **O**, **P**, **Q**, **R** and **S** classes based on their structural features (Figure 12)) reveals information on how different functional groups and the structural features present in esterquats affect their antimicrobial properties. The analysis of the collected data presented in Table 8 revealed that the length of the alkyl chain has a significant effect on the toxicity of the tested esterquats, which was found to increase with the elongation of the chain up to a certain amount of carbon atoms (usually between C_12_–C_14_) [18,107]. It is hypothesized that the presence of long alkyl groups contributes to the high hydrophobicity of the cation and facilitates its sorption onto cells, which ultimately results in the disruption of cellular membranes [135]. Further elongation of the alkyl chain usually decreases the solubility of the tested compound in water and results in lowering its toxicity [107,136]. This phenomenon was first noted for conventional QASs and described repeatedly as a ‘cut-off’ effect [137].

The size of the spacer between the ammonium groups might also exhibit a notable effect on activity toward various microbes. Thus, the gemini-type esterquat with two methylene groups between the oxycarbonyl group and the ammonium group at the linker moiety showed slightly lower activities (expressed as MIC) compared to the compound of similar structure with three methylene groups (Table 8, Entry 27: 22 µg/mL vs. Entry 28: 10 µg/mL, against *E. coli*) [18]. 

A direct comparison of the values noted for esterquats with their alkyl analogs, such as *N*,*N′*-bis(dodecyldimethyl)-1,2-ethanediammonium dibromide or benzalkonium chloride showed a 10- to 20-fold decrease in activity in the case of compounds comprising an ester group [107]. This is illustrated by the MIC values against *E.coli*, where esterquats exhibited MIC values above 10 µg/mL (e.g., Table 8, Entry 82: 11.7 µg/mL and Entry 90: >28.7 µg/mL), compared to as low as 0.62 µg/mL for *N*,*N′*-bis(dodecyldimethyl)-1,2-ethanediammonium dibromide and 2.6 µg/mL for benzalkonium chloride [107].

Interestingly, it was also established that Gram-negative bacteria exhibited significantly greater resistance to esterquats with chloride, bromide and iodide anions than Gram-positive bacteria. Thus, MIC values for Gram-negative *E. coli* were found to be 400 µg/mL (Table 8, Entry 29 and Entry 33) whereas for Gram-positive *S. aureus* the value was determined to be 200 and 5 µg/mL, respectively (Table 8, Entries 29 and 33).

The different bactericidal activities of esterquats can be explained by the absence of an outer membrane and the occurrence of negatively charged teichoic acid molecules within a thick peptidoglycan layer on the surface of *S. aureus*. This makes the surface of Gram-positive bacterial cell membranes more attractive to positively charged esterquats and more likely to be damaged compared to Gram-negative representatives. Moreover, the presence of a number of small channels of porins within the outer membrane of Gram-negative bacteria may be helpful in blocking the entrance of quaternary ammonium compounds into the bacterial cell, making Gram-negative bacteria more resistant [135].

In the case of esterquats comprising an herbicidally active anion in their structure, the effect of increasing toxicity with alkyl chain length as well as the subsequent cut-off effect (usually for alkyls exceeding C_12_–C_14_) was also noted. Thus, MIC values for esterquats with different alkyl chain length followed the order C_14_H_29_ >> C_10_H_21_ > C_12_H_25_ against *E.coli* (33.8 >> 15.5 > 8.1 µg/mL, as shown in Table 8, Entry 46, 44 and 45, respectively) [47].

It was also established that the toxicity of esterquats with herbicidal anions is higher by two or three orders of magnitude compared to conventional herbicides (e.g., Table 8, Entry 47-*E. coli*: EC_50_ 16.3 µg/mL, (µg/mL) compared to EC_50_ of dicamba + 2,4-D acids found to be 5.5 µg/mL [47]. Further analyses performed with other herbicides indicated a similar trend and the high toxicity of the studied esterquats has been associated with the significant toxicity exhibited by the amphiphilic cation used in the synthesis [47].

Analysis of the data collected from various studies regarding esterquats revealed how the presence of different functional groups in the cation may affect their toxicity. First, it was confirmed that the length of the alkyl chain has a crucial influence on the functioning and development of microorganisms, particularly bacteria and fungi. Moreover, the distance between the oxycarbonyl group and the nitrogen atom in the linker of gemini-type esterquats may also contribute to increased toxicity. It was also confirmed that esterquats exhibit a slightly lower toxic effect on microorganisms compared to their analogs deprived of an ester bond; however, further research is still required in this field. In the case of esterquats with an herbicidal anion, it was established that a high level of toxicity is mainly caused by the cationic part of esterquats rather than the herbicidal anion. A literature survey also revealed that data presented from different studies are inconsistent: different research groups utilize different tests as well as different bacterial strains to evaluate toxicity, or the obtained results are being presented with the use of different units. Therefore, the standardization of protocols and guidelines is required to ease the comparison of collected results from various experiments, but due to hydrolysis toxicity can decrease by orders of magnitude after a couple of days in a water environment [133].

#### 4.2.2. Toxicity of Esterquats toward Human Cells

In the context of the above results indicating that esterquats can have a significant effect on cellular function, the question should be asked whether they can also induce hemolysis or pose safety concerns for human cells. Unfortunately, there is no sufficient amount of data undoubtedly indicating that these groups of compounds are practically safe for humans in direct contact. Nonetheless, Para et al., established that there was no significant increase in necrotic cells in either control or surfactant-containing wells at concentrations between 0.01 and 0.1 mM, thus suggesting that both tested surfactants (choline dodecylate esters with bromide anion) with or without lysozyme are safe for application on human cells at those concentrations [6]. Furthermore, other research indicates that the addition of suitable groups, e.g., hydroxyl or methyl ester at the end of the alkyl chain of QASs greatly reduces their cytotoxicity [138]. However, in light of the potential applications of esterquats as a replacement for other surface-active QASs, it is strongly recommended to fill the gap and provide more data regarding the potential impact of these compounds on human health.

#### 4.2.3. Toxicity of Esterquats toward Mammalian and Aquatic Life

There is still scarce information regarding the toxicity of esterquats for other organisms, such as mammals, fish, shellfish, crustaceans, or algae. However, the available data summarized in Figure 13 and Table 9 (detailed structures are given in Appendix A) [12,21,111,133] indicate that they generally follow the same trends as conventional QASs [134], but are usually less toxic. Commercially available esterquats, TEAQ and DEEDMAC (Table 9, entries 1 and 3) were thoroughly evaluated and all data about their ecotoxicity are available in the ECHA database [139,140]. TEAQ esterquat was not classified as “hazardous to the aquatic environment”; however, DEEDMAC was classified as “Harmful to aquatic life with long lasting effects”.

It should be noted that hydrolysis of esterquats affects the determination of their toxicity in water systems. A recent study of alkyl betainate salts revealed that esterquats solution in a test medium can rapidly lose their toxic properties in a matter of days [133]. It is more complicated by the fact, that the stability of esterquats is influenced by various factors such as salinity, temperature, and pH of the test medium. Consequently, even minor alterations in experimental conditions can significantly impact the determined parameter values.

Following the presented results (i.e., biodegradability), it is certainly true that some esterquats are better alternatives to classical tetraalkylammonium salts, and are widely used in the chemical industry. However, despite their industrial popularity, it is important to encourage continued caution and further evaluation for both the esterquats themselves and the products of their hydrolysis and metabolic transformations.

## 5. Applications

### 5.1. Surfactants, Emulsifiers and Foam Stabilizers

The largest areas for applications of esterquats are as surfactants, emulsifiers and foam stabilizers (examples are provided in Figure 14, structures of the types **A**, **B**, and **C**, respectively).

The literature survey confirms the potential of using esterquats as biodegradable surfactants that can be produced from raw materials of natural origin as part of a circular economy. For example, amphiphilic esterquats made from glycine betaine have been used to obtain cationic surfactants with readily biodegradable anions derived from methanesulfonic acid (Figure 14, structures of type **A**) [21,116].

For surfactants, the main physicochemical parameter is a critical micelle concentration (CMC) that defines their surface activity and self-assembled aggregation. The overview of CMC values for the esterquats **A** confirms their effectiveness as surfactants (CMC 0.1–1.2 mmol/L) and demonstrates dependence on the length of alkyl chains [116].

Among the synthesized organic salts, two compounds containing 18 carbon atoms in the alkyl chain of the cation (Figure 14 **A**, R_1_ = C_18_H_35_ and C_18_H_37_) deserve special attention. The first one was characterized by excellent emulsifying properties, due to the presence of a double bond in the aforementioned chain. The subsequent tests qualified this compound as a readily biodegradable substance (reaching 60% degradation within 28 days), suggesting it to be an extremely promising candidate for use in industry. When this compound was tested for its susceptibility to hydrolysis, it was established that the compound is stable under acidic conditions and exhibits a high degree of decomposition under alkaline pH [116]. This important feature allows for the possible control of the rate of decay of said esterquat and thus minimizes the risk of its potential accumulation in the environment.

The stability of the gemini surfactants containing either a three-carbon or a six-carbon spacer (Figure 14, structures of the type **B**, n = 1, R_2_ = CH_2_ or C_4_H_8_) was compared with their monomeric counterparts (Figure 14, structures of the type **C**). The results indicated that mono-esterquats are less susceptible to alkaline hydrolysis than their gemini analogs. Intriguingly, a gemini surfactant containing a C_9_ alkyl chain and a carbon atom spacer demonstrated the greatest efficiency in its application as a foam stabilizer. Its degree of decomposition under the influence of biotic factors did not allow for its recognition as readily biodegradable, as opposed to the similar monomeric derivative [19]. The team of Tehrani-Bagha et al. also studied susceptibility to biodegradation of this type of compound which was established to be lower for gemini-type compounds (60% biodegradation was achieved after 35–40 days) than for the similar monomeric derivatives. Nonetheless, taking into account the results of decomposition under the influence of biotic factors and the excellent surface activity parameters, the possibility of implementing the discussed compounds as commercial surfactants is highly probable [3].

Another potential industrial application was demonstrated recently with the use of an interesting group of diester-bonded cationic gemini surfactants (Figure 14 **D**, R_1_ = C_12_H_25_, C_14_H_29_ and C_16_H_33_). The compounds tested in this study were found to possess excellent surface-active properties and low critical micelle concentration (CMC) values. Additionally, the collected results revealed their significant antibacterial activity against Gram-positive bacteria. Undoubtedly, these findings open up a new path toward multifunctional compounds, such as cleaning agents enriched with targeted antibacterial (disinfectants) or cytotoxic activity (cytostatic drugs) [20].

Appendix A contains the molar masses of the chosen esterquats, their CMC values, and surface tension at CMC. Generally, CMC values vary from 0.00098 g/L for gemini esterquats to 18.5 g/L for monomeric derivatives (Appendix A) [15,20]. The influence of the structure of esterquats on CMC values is analogous to typical QASs, which means that the elongation of the alkyl chain results in the lowering of this parameter. Gemini compounds are characterized by lower values compared to those of monomeric compounds comprising the same alkyl chain in the structure of the cation. Interestingly, the structure of the anion can significantly affect the CMC as well; for example, the exchange of bromide anions to chloride results in decreasing CMC values orders of magnitude (Appendix A, Entries 25 and 27 as examples for choline-type monomeric esterquats and 135–137 and 139–141 or 131 and 139 for betaine-type gemini esterquats). It should also be emphasized that the elongation of the spacer between quaternary nitrogen atoms in gemini esterquats was found to marginally affect their surface properties (Appendix A, Entries 7 and 10, 31–38 for choline type esterquats, and 104–112 for betaine-type esterquats).

Also important for surfactants is their foam properties (e.g., foam stability, foam expansion ratio), as they have a significant impact on their practical application; therefore, these parameters were also studied. Foam expansion is defined as the ratio of the volume of foam formed to the volume of liquor (liquid) used to generate the foam whereas foam stability is the time that foam will maintain its initial properties as generated. The foam height and foam stability of several cationic gemini surfactants (Appendix A, Entries 66–74) are summarized in Figure 15. It was established that as the length of the hydrophobic chain increases, the surfactants become more abundant in foam, and the foam stability increases. Additionally, compound ‘*h*’ in Figure 15 showed the best foam height compared to cationic gemini surfactants that have other spacer lengths, but its foam stability was found to be relatively poor [18]. 

Numerous scientific patents disclose the use of esterquats as components of creams and lotions, shampoos, conditioners, or various other skin cosmetics. Besides improving their stability, esterquats have also been proposed as a safer replacement for non-natural origin ingredients such as silicones. Due to the rising awareness of the customers and their growing requirements, this action is part of the modern marketing campaign promoting eco-friendliness [38,39,141,142]. The esterquats possess various functions in the formulations, acting as emulsifiers (for example Henkel’s TEAQs or Colgate Palmolive’s DEEDMACs presented in Figure 2), potentially also as fragrance carrier or precursor, or intensifiers of other compounds’ effects.

The potential to facilitate dispersion of solid cosmetic ingredients in lower temperatures or remaining in liquid state at room temperature can also be considered as one of their most important functions, as it may reduce energy and solvent consumption [39,40,141,142,143,144]. Silicone-based compositions can obviously provide desired properties, such as smooth and soft feeling, although, analogously as in the case of the conventional QASs, petrochemical emollients, or mineral oils, huge efforts are being put into reducing their use. Thus, products derived from natural substances are being increasingly explored and utilized in products, even despite providing less beneficial effects compared to more toxic, old representatives. For this reason, there is still a large scientific gap, which impels the need for future improvement and development of products containing esterquats as replacements for other chemicals [37,39,142].

In the case of hair care products, compositions containing esterquats were also proven to be used in conditioners, as detangling and antistatic agents, leaving hair easier to comb. Such compositions can also provide benefits in lengthening the styling hold and frizz control. Additionally, some esterquats provide increased hair shining, flexibility and elasticity. They are used particularly in combinations with betaine, non-esterquat cationic surfactants, specific amidoamines (monoamides of alkyldiamines), or glycerin esters [37,39,40,143,144].

### 5.2. Flocculants, Flotation Collectors and Fabric Softeners

An interesting application of unsaturated-functionalized esterquats was achieved by their polymerization. In effect, a number of compounds were successfully obtained, the cation is shown in Figure 16 (**A**), and the anions were as follows: benzoate, salicylate, acesulfame and saccharinate. The synthesized compounds were effective as flocculants of negatively charged particles, e.g., in the case of various yeast suspensions. The tested esterquats were more efficient than the applied reference cationic polymers without an ester bond. It should be noted that the newly reported functionalized advanced polymeric materials have also been classified as a unique group of poly(ionic liquids) (PILs). It is noteworthy that their synthesis did not require the use of solvents (except for the anion exchange stage), purification, or drying, which is an inherent step in methods of preparation of “standard” flocculants [22].

Flotation is an industrially important technique that relies on differences in surface properties of different minerals to achieve separation of hydrophobic materials from hydrophilic. A novel quaternary ammonium surfactant called M-302, shown in Figure 16 (**B**, n = 4), containing ester bonds and hydrocarbon tails was proposed as a cationic collector for iron ore flotation. Generally, the flotation collectors are naturally attracted to metallic surfaces making them more hydrophobic. However, collected results revealed that M-302 tended to adsorb on the surfaces of quartz more than that of hematite, and in effect, their separation was significantly facilitated [35]. At the condition of natural pulp pH (7.38), the starch dosage of 8.0 mg/L, and the M-302 dosage of 20.0 mg/L, quartz and hematite could be separated effectively with 93% recovery. Interestingly M-302 is a readily biodegradable compound, which promotes consideration of its application on a commercial scale.

Introduced in the early 1980s as a response to potentially environmentally harmful fabric rinsing agents, esterquats are also being widely used in the textile industry as fabric softeners. They improve the quality of materials, making them more pleasant to touch and smell. In addition, they affect the overall strength and durability of textiles. Because of the presence of easily cleavable ester bonds in their structure, esterquats are more susceptible to degradation, which minimizes the risk of skin allergies or other dermal issues. For this reason, esterquats are excellent fabric rinsing alternatives that meet more stringent requirements established as a result of intensified pro-ecological initiatives [1,29].

### 5.3. Pharmaceuticals and Antiseptics

The pharmaceutical industry is one of the most promising areas for novel and innovative applications of esterquats. These compounds exhibit various biological activities, while the presence of labile ester bonds makes them attractive candidates for drug delivery. Thus, gemini surfactants derived from choline (Figure 17, structures of type **A**) are effective biocides. Data on biocidal activity showed that the elongation of the alkyl chain of the surfactant enhances this feature [74].

Betaine ester-shell functionalized hyperbranched polyethylenimines comprising multiple ester bonds (Figure 17, structures of type **B**) are not only highly antibacterial but are able to effectively store large amounts of antibiotics and antiseptics. Breaking down under weakly alkaline conditions (pH = 7–9) allows the controlled release of actives trapped inside the base-labile betaine ester shell. Moreover, these ester-shell functionalized hyperbranched polyethylenimines obtained from long-chain alkyl bromoacetate proved to be compatible with organic resins and could be employed in the production of antibacterial fibers and coatings [17].

In a study performed by Lundberg et al., surface-active betaine esters (Figure 17, structures of type **C**, where n = 1, R_1_ = C_10_H_21_, C_12_H_25_, C_14_H_29_, C_18_H_35_) have been proposed as candidates for the implementation in the field of pharmaceutical applications brought about by the presence of a readily hydrolyzable linkage in their structure and the formation of harmless degradation products. In the framework of the research, the phase behavior of two systems was investigated: dodecyl betainate-dodecanol-betaine hydrochloride-D_2_O and dodecyl betainate-phosphatidylcholine (PC)-ethanol-D_2_O. The results presented in this study indicated that the cationic betaine esters derived from fatty alcohols are a class of compounds possessing both hydrophobic and hydrophilic portions with potential use in drug delivery, particularly when formulated with PC or other lipid excipients [23]. Additionally, structurally similar gemini hexylbetainates were used to increase the availability of the antidepression drug, imipramine through the formation of micelles [145].

According to another report, appropriately designed derivatives of QASs and lysosomotropic substances (QDLS) (Figure 17, structures of type **D**) simultaneously demonstrate surface active properties and biological activity. Because living organisms constitute an electrolyte-rich environment which, in turn, strongly affects the properties of QDLS, the aim of this research was to analyze their adsorption processes in air-electrolyte systems. The sources of electrolytes were various sodium salts: NaCl, NaBr, and NaClO_3_, and the measurements were carried out at two temperatures: 21.0 °C and 36.6 °C. It was concluded that the addition of salt along with the increase in temperature contributes to an improvement in the surface activity and accelerates the process of QDLS adsorption at the air-electrolyte interface. However, it should be noted that the observed enhancement in adsorption properties depended on both the type of utilized anion and its concentration [24].

A thorough analysis of a series of monocationic betaine-based esterquats (Figure 17, structures of type **E**), revealed their interesting properties originating from the presence of an ester bond. Their antimicrobial effect as well as the rate of their hydrolysis in aqueous solution increased upon elevation of pH. Moreover, the bactericidal activity of such compounds increased with the length of the alkyl chain up to 18 carbon atoms. Bearing in mind that the products of hydrolysis are human metabolites, these esterquats could be employed as disinfectants, and successfully utilized not only in the food industry but also directly on the surface of the human body [16].

Recently, a direct relationship between the structure of dicationic surface active esterquats (Figure 17, structures of type **F** and **G**) and their biological activity was analyzed in experiments, wherein erythrocytes were utilized as a simple model of the biological membrane. The results showed that depending on the applied concentration as well as the esterquat chemical structure, the osmotic resistance of erythrocytes can be influenced in different ways. At sufficiently high concentrations, the membrane is destroyed, ultimately causing the breakdown of red blood cells. Surfactants containing 10, 12 and 14 carbon atoms in the alkyl chain and structurally larger linkers were characterized by higher activity. This feature may become useful in designing the structure of new compounds with a pharmaceutically desired activity [25].

Interestingly, as shown in Figure 18, glycine betaine can also be used in clean technology as the finishing reagent for the preparation of antibacterial fabrics. In this unique approach, the reactive carboxyl group of betaine binds to the cellulosic fabrics via esterification, while the quaternary ammonium moiety exerts an antibacterial effect. The obtained fabric exhibited excellent antibacterial and anti-serum protein adsorption capacities without compromising its wearing comfort properties, such as flexibility or vapor transmissibility [10].

### 5.4. Agrochemicals

In the area exploring the subject of agrochemicals, a group of choline-type QASs that possess an ester bond in their structure and anions derived from herbicides belonging to the group of synthetic auxins (2,4-D, MCPA, MCPP and dicamba) (Figure 19, structures of type **A** and **B**) were successfully synthesized and characterized. Such esterquats possess generally high thermal stability over a wide temperature range. In effect, the potential mobility of the substances and their drift to neighboring agricultural lands is significantly reduced compared to esterified forms of the analogous herbicides, which are deprived of the ionic bond. Performed field and greenhouse tests demonstrated that some of these new forms of herbicides exhibited high activity against dicotyledonous weeds, in some cases exceeding the effectiveness of commercial preparations. Due to their notably high efficiency, they can be applied at much lower doses, which additionally contributes to the minimization of the risk of environmental pollution [12].

Interestingly, the ester bond was successfully utilized for the synthesis of functionalized esterquats, which possess a biologically active moiety incorporated intentionally into the cation (Figure 19, structures of type **C** is an example of an esterquat with an incorporated MCPA structure into cation). In 2016, Piotrowska et al. described esterquats, in which choline cations were linked with two selective herbicides (2,4-D and MCPA) [26]. The design of such unique structures originates from previous reports revealing that ester forms of such phenoxy acids can be much more active than their non-esterified counterparts. However, they exhibited relatively high volatility, established as their greatest weakness. This weakness was eliminated when esterquats were used in formulations, mainly due to the presence of ionic bonds that are responsible for the substantial reduction in vapor pressure [13]. This concept was further developed by the use of ion exchange reactions, wherein choline-type cations functionalized with appropriate herbicides were combined with structurally identical or other herbicidally active anions. In effect, the obtained new active forms exhibited the potential for synergy or a broadened spectrum of activity, which in turn, also is able to minimize the escalating current phenomenon of herbicide-resistant weeds [9].

Recently, alkyl betainate cations (Figure 19, structures of type **D**) were combined with an anion exhibiting herbicidal activity, commercially known as iodosulfuron methyl. It was proven that these esterquats are characterized by a lower potential for bioaccumulation or migration to groundwater than the commercial preparation of this herbicide. The subsequent greenhouse tests performed on the winter oilseed rape at an extremely low dose of active ingredient (10 g ha^−1^) confirmed the excellent biological activity of the esterquats, which was statistically comparable to that of the commercial preparation as a reference. These findings make them an attractive, more environmentally friendly, replacement for known forms of sulfonylurea-based herbicides [11].

In 2022, Stachowiak et al. proved that betaine-based esterquats (Figure 19, structures of type **D**) comprising bromide or dicamba anions show phytotoxic effects toward white mustard in a stage of 4–6 leaves. However, it should be noted that the plant growth-inhibiting effect depended on the length of the alkyl substituent, and in some cases, the esterquats used outperformed applied reference commercial substances. Moreover, newly synthesized compounds presented high herbicidal activity without the utilization of any other additives. This fact is important in light of the search for substitutes for commercial herbicides relying on the use of biodegradable materials of natural origin. Additionally, the authors of this research established that the cation in esterquats should not be regarded only as an inert surface-active agent but rather as the moiety that can substantially influence the development of terrestrial plants. However, as shown in Figure 20, the biological response is highly dependent on the length of the alkyl chain in the cation [14].

### 5.5. Materials in Electrochemistry

Chami et al. also proposed the application of esterquats in the field of electrochemistry, specifically in Figure 21, structures of type **A** (n = 2, R_1_ = C_11_H_23_ R_2_ = CH_2_) and **B** (n = 2, R_1_ = C_11_H_23_ R_2_= C_12_H_25_). The evaluation of the results obtained after examining their behavior with various electrochemical techniques, surface characterization and quantum chemistry calculations allowed them to establish their positive effect on the process of iron corrosion inhibition in 1 M hydrochloric acid solution. The Tafel curves indicated a decrease in the corrosion current density with the addition of different concentrations of both inhibitors. However, it has been proven that gemini-compounds (Figure 21, structures of type **A**) are more efficient than the analogous monocationic compounds (Figure 21, structures of type **B**). Studies of electrochemical impedance spectroscopy (EIS) have also shown that the compounds inhibit iron corrosion in the process of adsorption of surfactant molecules on the iron surface. Experimental studies confirmed the validity of theoretical considerations and confirmed that compounds **A** are good corrosion inhibitors [46].

Recently, biodegradable betaine-based aprotic ionic liquids from renewable natural resources (Figure 21, structures of type **C**) have found application as effective absorbents for SO_2_. The study showed that one of them, containing an alkyl chain consisting of four carbon atoms, can maintain a high absorption capacity and rapid absorption rates for even 25 cycles. The electrochemical windows of the betaine-based ILs used in this paper were evaluated by cyclic voltammetry on a glassy-carbon electrode. Evidently, all of the measured ILs possessed relatively wide electrochemical windows, ranging up to 2.9–4.3 V [146].

The quaternary betaine-type ammonium salt (Figure 21, structures of type **C**) on AA2024-T3 in 0.01 mol/L NaOH solution was investigated using, among others, the weight loss method (WLM) and electrochemical impedance spectroscopy (EIS). Interestingly, the tested molecules proved to be effective corrosion inhibitors achieving high protection efficacy (70–99%). In addition, the analyzed compound was proven to be a mixed-type inhibitor that mainly inhibits the anodic corrosion reaction [147].

LiFePO_4_ batteries, mainly used in hybrid and electric cars, are a longer-lasting and safer variant of lithium-ion batteries. In addition to high current efficiency, they are also characterized by a long service life. Studies performed by Pan et al. focused on the search for electrolytes for such batteries to increase their stability and safety. In effect, the designed and newly obtained piperidinium-based ester-functionalized ionic liquid, characterized by a low melting point, low viscosity and good thermal stability, enabled the achievement of an electrochemical window as wide as 6 V. Additionally, the tested system also demonstrated excellent cycle efficiency, which undoubtedly proves that other esterquats, such as choline-type or betaine-type esterquats, exhibit excellent potential to be successfully used as electrolytes in lithium-ion batteries in the near future [27].

## 6. Conclusions and Perspectives

Many years of intensive studies on choline-type and betaine-type esterquats, as unique representatives of quaternary ammonium salts (QASs), have allowed us to gather a comprehensive database of well-established synthesis methods characterized by acceptably high yields and purities. In the past, the majority of esterquats were synthesized via the esterification of aminoalcohols with fatty acids followed by quaternization [9,55,61,64,68]. Nonetheless, other routes of their synthesis are still being developed, in which other naturally derived substrates are used, such as glycine betaine (*N*,*N*,*N*-trimethylglycine) [10,11,88] or carnitine (3-hydroxy-4-(trimethylammonio)butanoate) [56,88]. It was repeatedly confirmed that their excellent tunability also enables the adjustment of various physicochemical properties, such as hydrophobicity and surface activity, which is an extraordinarily advantageous feature in terms of their plausible application.

The structural similarity of esterquats to some beneficial compounds widely present in living organisms (such as choline and betaine) allows for considering them as potentially more environmentally friendly alternatives to fully synthetic QASs. Moreover, the presence of cleavable ester bonds within their structure facilitates decomposition into compounds deprived of amphiphilic properties that have been recognized as a main contributor to the toxic effect of QASs. In effect, the issue associated with the use of some hazardous commercial cationic surfactants or disinfectants can be successfully minimized by simple replacement of active ingredients with their ‘greener’ analogs.

Undoubtedly, choline- and betaine-type esterquats constitute a group of chemicals that will inspire scientists for many decades. It should be stressed that the idea of incorporating an active substance into the cation of esterquat has extreme potential to be successfully applied in the near future in medicine, particularly in the development of modern drug delivery systems, which will be able to release a drug (e.g., antibiotics or antimicrobial agents) in a controlled way. The market of agrochemicals would also gain enormous benefits if, e.g., plant growth stimulators (such as indole-3-acetic acid (IAA) and indole-3-butyric acid (IBA)), responsible for significant enhancement of cultivated plant development, were utilized in this concept. One should also expect an increase in the interest in esterquats composed of a betaine-type cation, which for years have been overshadowed by compounds composed of a cation derived from choline. However, as a cheap and environmentally friendly raw material of natural origin, betaine has recently gained the interest of the scientific community, which has directly led to an increase in the number of innovative scientific works on the synthesis and applications of various derivatives of this compound, including esterquats. The example of effective functionalization of the surface of cotton fabric with glycine betaine demonstrates an insightful approach to designing novel, innovative materials with excellent additional antibacterial activity without impairing their usability.

However, there are still some areas wherein esterquats need further development, especially when considering the expansion of the scope of their commercial applications. First, it should be noted that many scientists deserve attention for their ingenuity in developing effective methods of obtaining esterquats. However, the majority of elaborate procedures are extremely complex or require some specific conditions, which makes them impossible to upscale to commercial production. Obviously, all new substances should be thoroughly tested to provide at least the most important basic parameters, such as solubility in water and surface activity as well as potential bioaccumulation in the environment. The question “What is the real relationship between the structure of esterquat and its ecological profile?” thus far remains an open question, which has not yet been solved. Unfortunately, there is a gap in the unification of the selections of physicochemical properties that are being determined for esterquats. Additionally, to date, biological tests (antimicrobial activity or toxicity) performed on esterquats have focused mainly on inconsistent experiments conducted according to different OECD protocols; hence, it is impossible to properly assess their environmental impact or draw additional unequivocal conclusions. Considering these drawbacks, future recommendations have been prepared. It is of the utmost importance to focus on the following aspects in the future:Simplification of synthetic procedures that comply with the concept of ‘green chemistry’ (e.g., utilization of nontoxic solvents, high atom economy, use of sources of natural origin, minimization of unit operations);Elucidation of standards for physicochemical characterization of newly synthesized esterquats. The proposed set should include tests focused on assessing their susceptibility to hydrolysis;Standardization of the toxicity assay is required to ease the comparison of collected results from various experiments. Esterquats should be assessed on an appropriately selected group of organisms with the use of one unified protocol or a group of protocols adapted to their respective application;The biodegradation of both ions of esterquats in soil as well as in water should be determined separately to understand their persistence in various environments. In effect, the standard activated sludge and bacteria isolated from soils should be used as a consistent set of experiments.

## Figures and Tables

**Figure 1 ijms-25-05761-f001:**
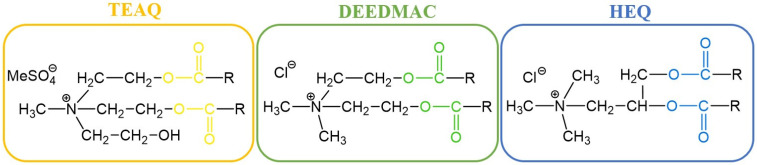
The structures of the most popular esterquats: triethanolamine esterquat (TEAQ), diethanol amine esterquat (DEEDMAC), and *N*,*N*,-dimethyl-3-aminopropane-1,2-diol esterquat (HEQ).

**Figure 2 ijms-25-05761-f002:**
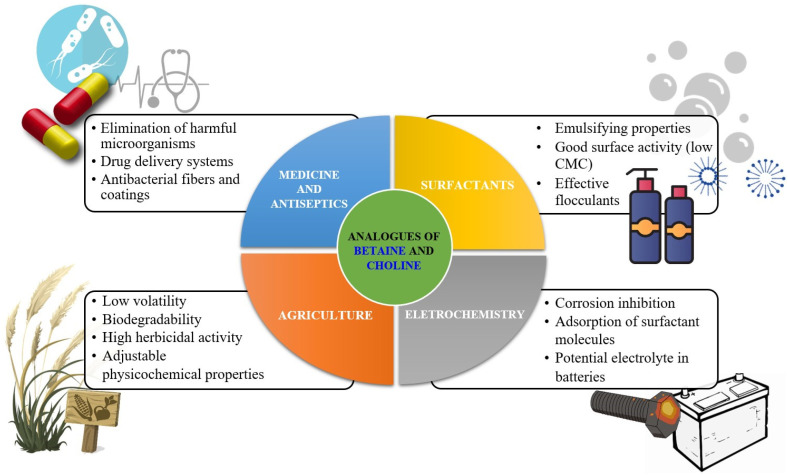
Reported applications of esterquats [3,9,10,11,12,13,14,15,16,17,18,19,20,21,22,23,24,25,26,27].

**Figure 3 ijms-25-05761-f003:**
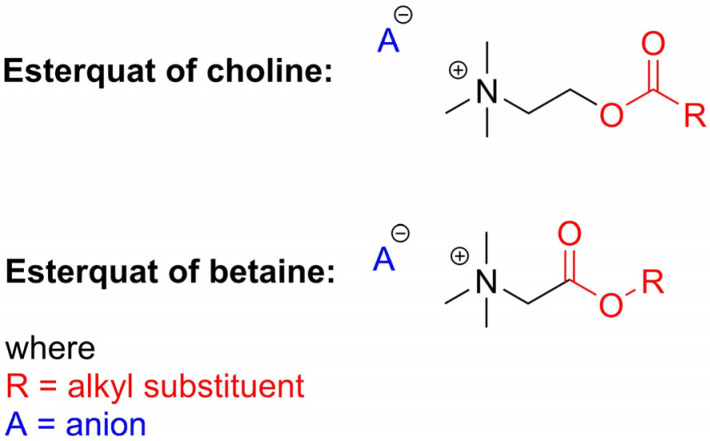
General formula of choline-type and betaine-type esterquats.

**Figure 4 ijms-25-05761-f004:**
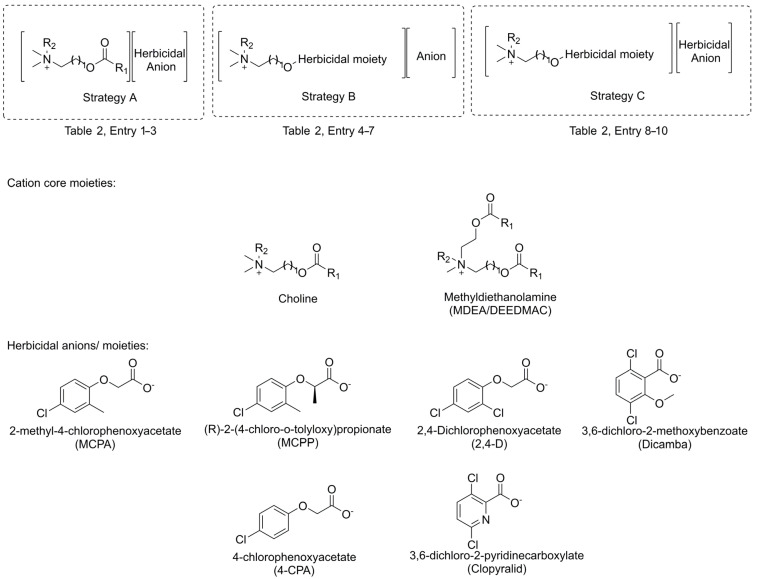
Derivatization strategies of known herbicides toward esterquats: (Strategy A) IL formation from the cholinium cation and herbicidal anion; (Strategy B) covalent modifications of the esterquat structure, in a manner similar to prodrug approach, with a formation of easily hydrolyzable ester bond; (Strategy C) a combination of a “prodrug” strategy and salt formation.

**Figure 5 ijms-25-05761-f005:**
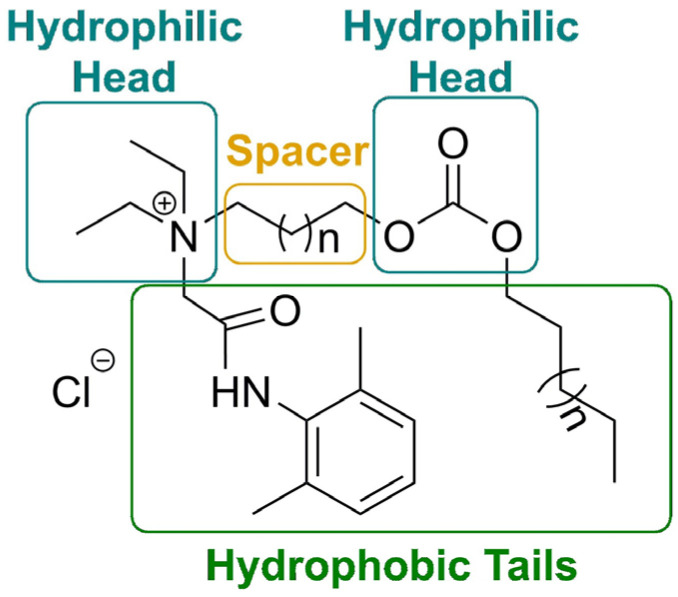
Structure of asymmetric gemini surfactant derived from carbonate and lidocaine moieties.

**Figure 6 ijms-25-05761-f006:**
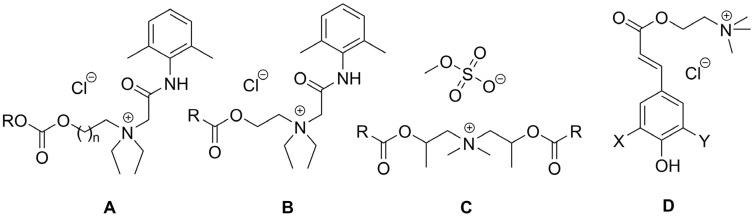
Left: structure of lidocaine-derived choline esterquat spontaneously self-assembling into Janus particles (**A**,**B**); middle: MDEA esterquat for delivery of cytostatic drugs and small RNA fragments (**C**); right: antioxidant sinapine esterquat derivative (**D**).

**Figure 7 ijms-25-05761-f007:**
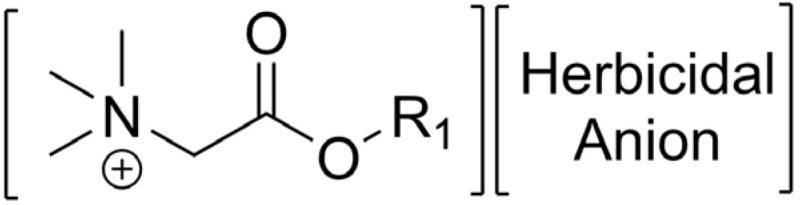
Structure of alkyl betainate cation paired with iodosulfuron methyl and dicamba anions.

**Figure 8 ijms-25-05761-f008:**
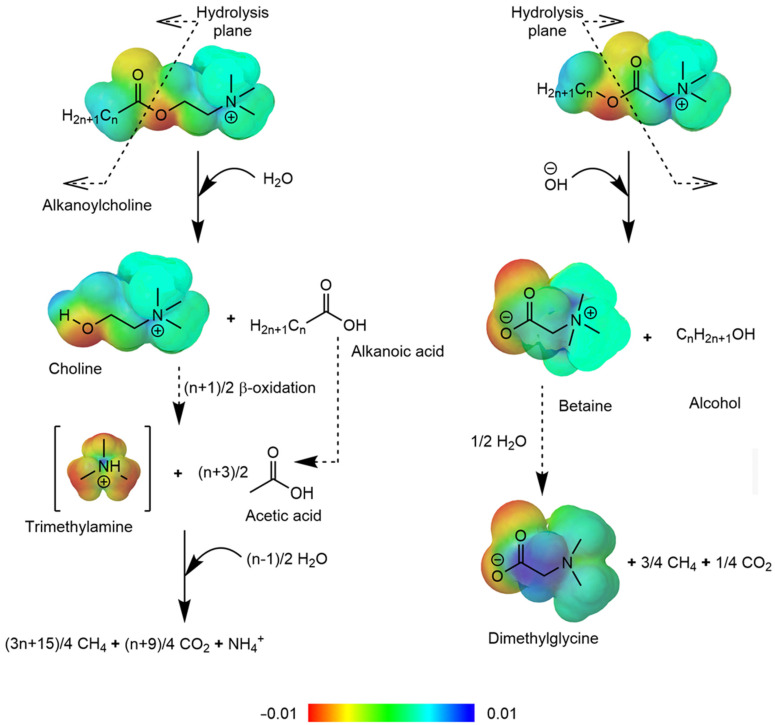
Proposed alkanoylcholine and alkylbetaine transformation pathways and stoichiometry under methanogenic conditions. The colored figures are total electron densities mapped as electrostatic potential between L0.01 and 0.01 e/au^3^, and the structures were modified to fit their potential surfaces [7,90]. The surfaces were drawn and optimized using Avogadro 1.2.0. and its basic optimization algorithm, and the potential surfaces were generated using Jmol ver. 14.32.77 (potential surfaces −0.1–0.1). Adapted from [7].

**Figure 9 ijms-25-05761-f009:**
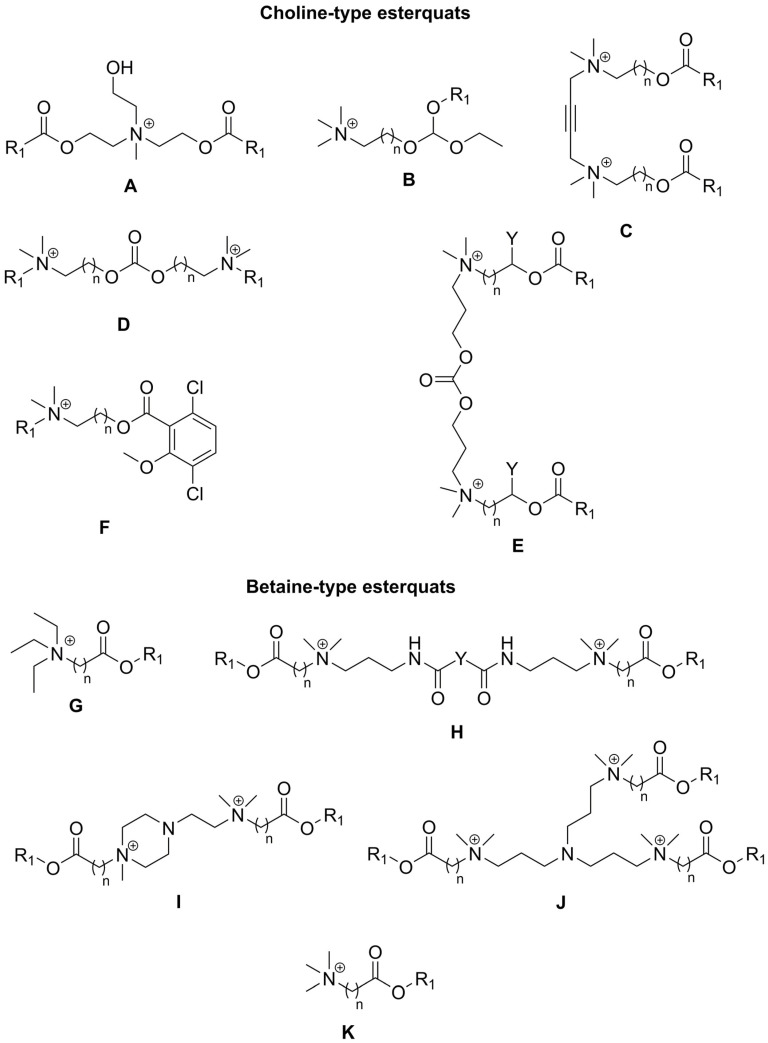
Structures of choline- and betaine-derived esterquats subjected to hydrolysis studies. Letters A–K indicate cation structure, for which data were presented in Table 6.

**Figure 10 ijms-25-05761-f010:**
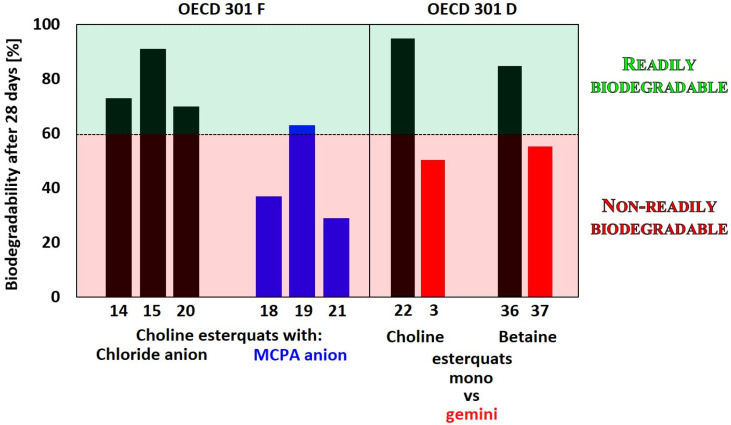
Biodegradation of esterquats and their classification (numbers of compounds correspond to entries from Table 7) [3,12].

**Figure 11 ijms-25-05761-f011:**
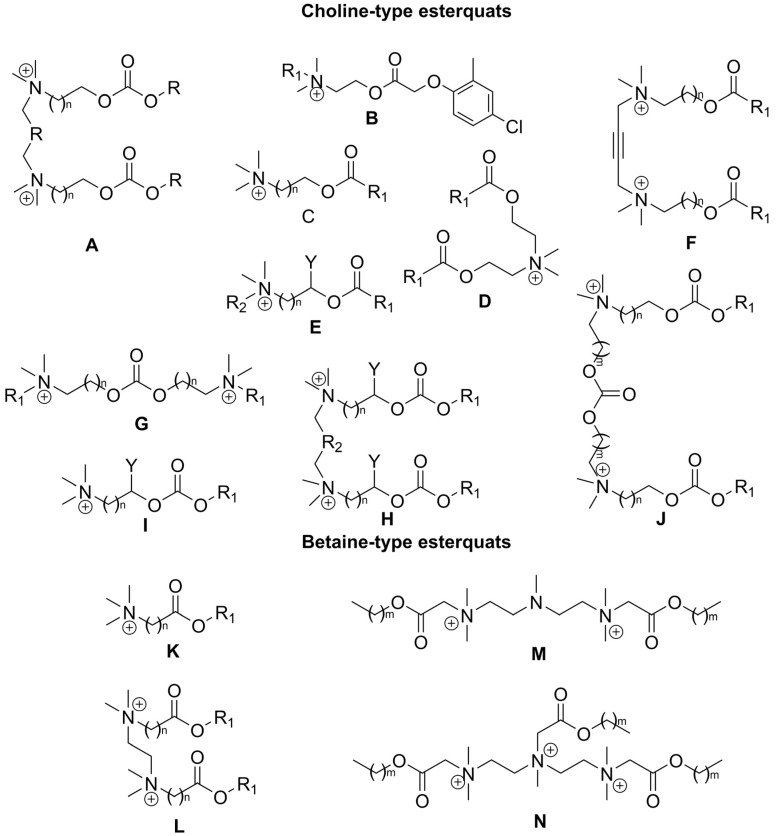
Structures of choline- and betaine-derived esterquats subjected to biodegradation studies.

**Figure 12 ijms-25-05761-f012:**
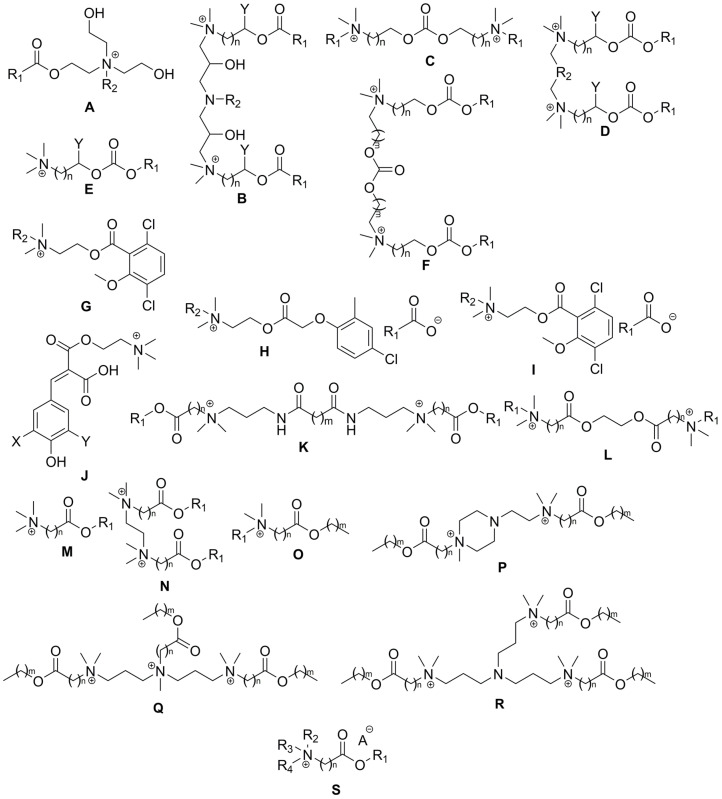
Structures of different esterquats subjected to antimicrobial activity studies.

**Figure 13 ijms-25-05761-f013:**
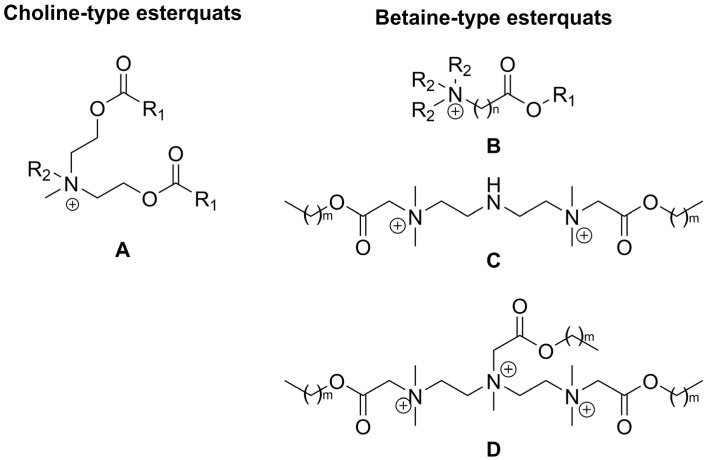
Structures of different esterquats subjected to toxicity studies. Letters A–D indicate cation structure, for which data was presented in Table 9.

**Figure 14 ijms-25-05761-f014:**
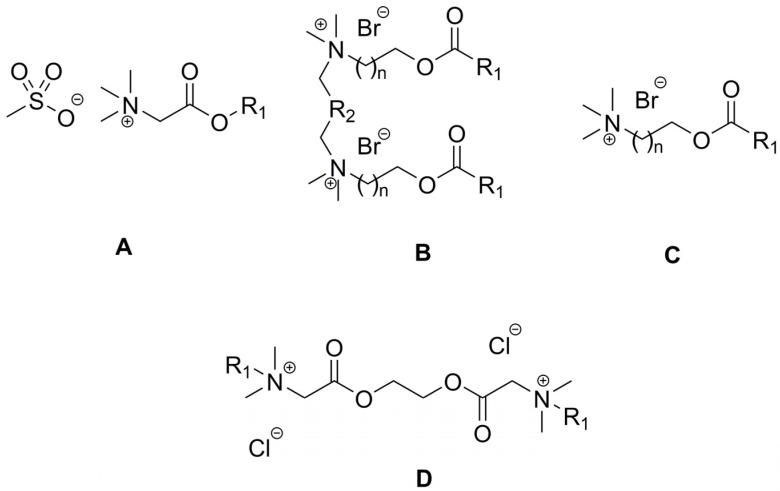
Examples of compounds utilized as surfactants, emulsifiers and foam stabilizers.

**Figure 15 ijms-25-05761-f015:**
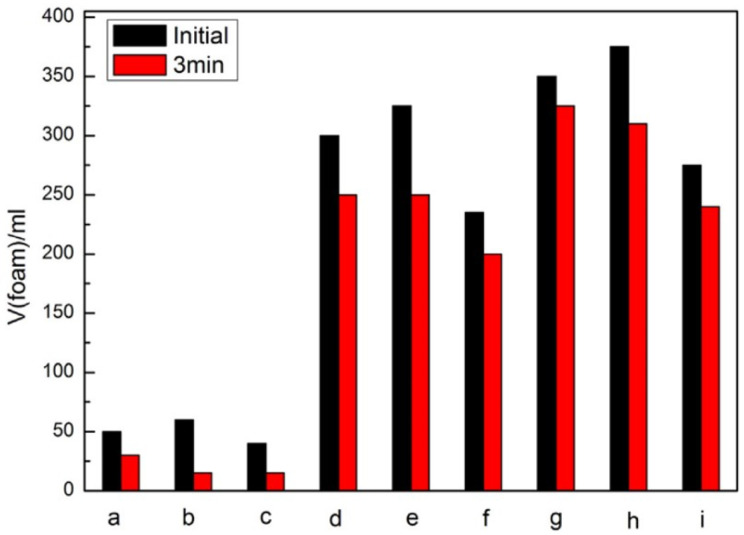
Plot of initial foam volume and after 3 min for 0.1 wt.% surfactants at 25 °C (cations structures are detailed in Appendix A, a: Entry 66, b: Entry 67, c: Entry 68, d: Entry 69, e: Entry 70, f: Entry 71, g: Entry 72, h: Entry 73, i: Entry 74). Reprinted from J. Mol. Liq. 299, 112248, J.Wu et al. “Cationic gemini surfactants containing both amide and ester groups: Synthesis, surface properties and antibacterial activity”, Copyright 2020, with permission from Elsevier [18].

**Figure 16 ijms-25-05761-f016:**
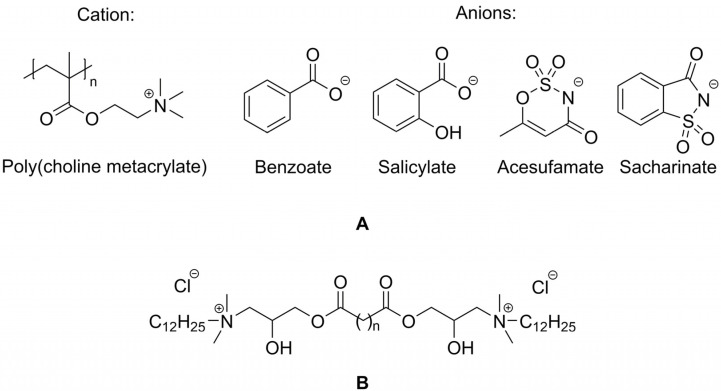
Examples of esterquats utilized as flocculants and fabric softeners.

**Figure 17 ijms-25-05761-f017:**
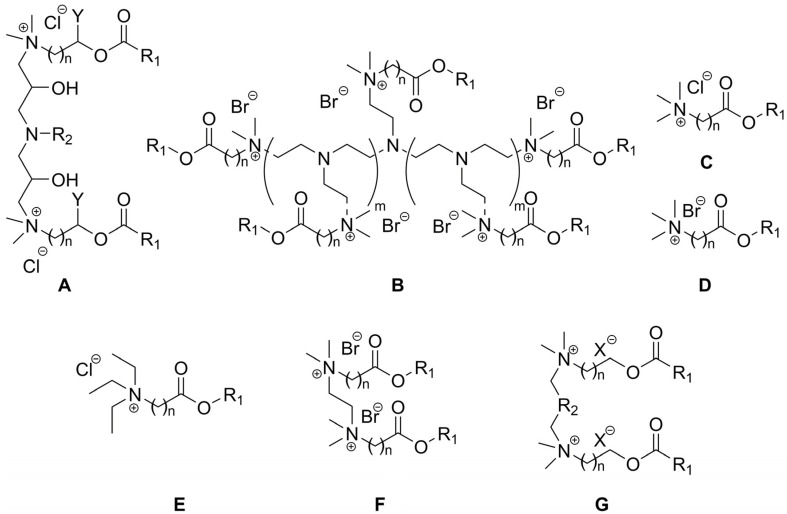
Example of compounds utilized as pharmaceuticals and antiseptics.

**Figure 18 ijms-25-05761-f018:**
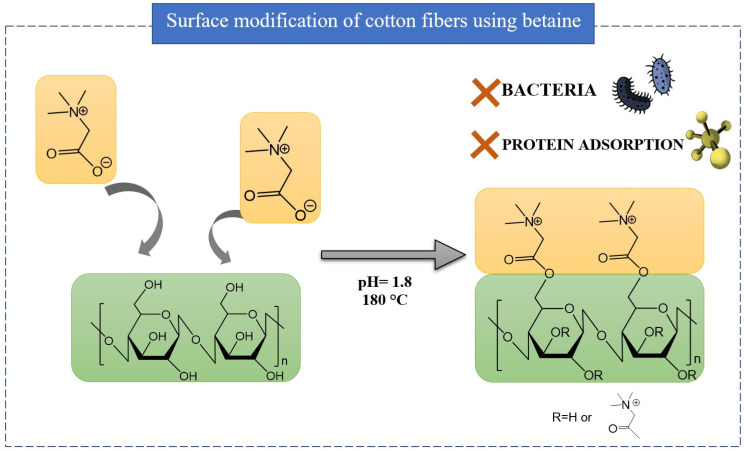
Scheme demonstrating the cotton fabric functionalized by glycine betaine [10].

**Figure 19 ijms-25-05761-f019:**
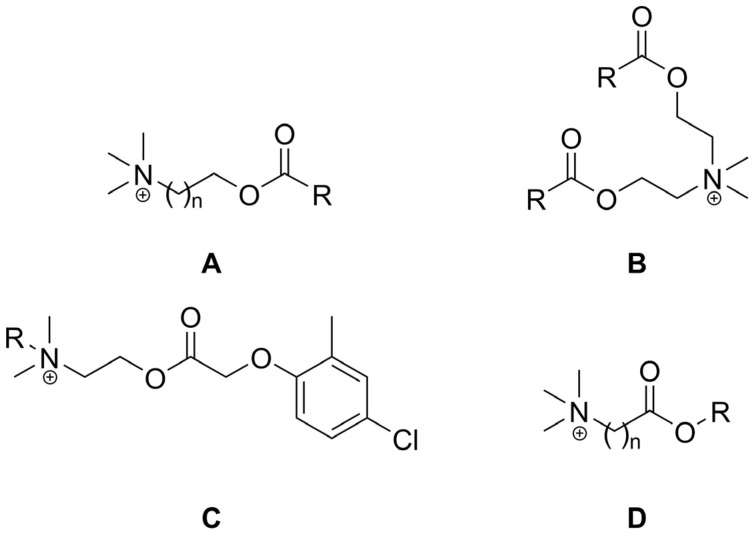
Examples of esterquat cations utilized as agrochemicals.

**Figure 20 ijms-25-05761-f020:**
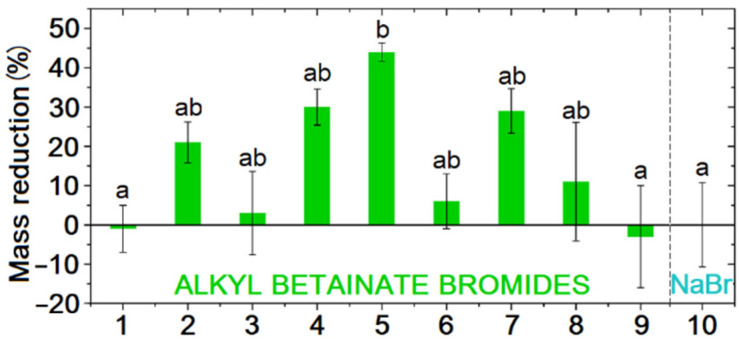
Phytotoxicity of alkyl betainate bromides (**D**, n = 1) toward white mustard, where the alkyl chain (R) starts from ethyl (**1**) to octadecyl (**9**) in comparison to sodium bromide (**10**). Letters above bars provide a visual representation of the statistical significance of the differences between the groups being compared [14].

**Figure 21 ijms-25-05761-f021:**
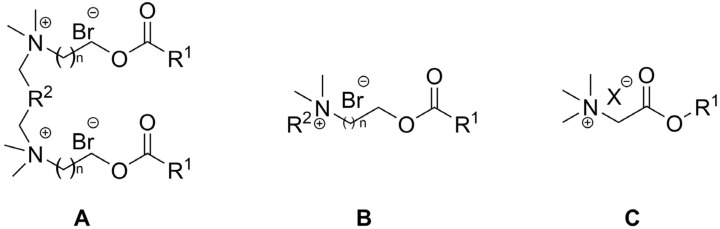
Example of esterquats utilized as materials in electrochemistry.

**Table 1 ijms-25-05761-t001:** The general methods of the synthetic methods for choline-based esterquats. Structures of esterquats are presented in Appendix A.

Synthetic Procedures	Location	References
Fischer esterification of aminoalcohols	Appendix A	[29,49,55,61,64,65,66,67,68,69,70,71]
Acylation and subsequent quaternization of amino alcohols	Appendix A	[3,6,9,47,63,72,73,74,75]
Other methods of derivatization of carboxylic acids	Appendix A	[56,57,62,76,77,78,79,80,81,82]
Utilization of biologically active compounds	Appendix A	[9,47,57,83,84]

**Table 2 ijms-25-05761-t002:** Reported herbicidal choline-type esterquats with biologically active moiety in their structure. Detailed structures of all cations are presented in Appendix A.

Strategy		Cation			Anion	Activity	Benefits	Ref.
		Core	R_1_ (Strategy A) or Herbicidal Moiety (Strategies B and C)	R_2_				
1	A	Choline	-CH=CH_2_	CH_3_	MCPA ^1^	Herbicidal	Replacement of conventional tetraalkylammonium cation in salt-type herbicides with more environmentally-friendly esterquat	[12]
	A	Choline	-CH=CH_2_	CH_3_	MCPP ^2^			
	A	Choline	-CH=CH_2_	CH_3_	2,4-D ^3^			
	A	Choline	-CH=CH_2_	CH_3_	Dicamba ^4^			
2	A	Choline	-C(CH_3_)=CH_2_	CH_3_	MCPA	Herbicidal		[12]
	A	Choline	-C(CH_3_)=CH_2_	CH_3_	MCPP			
	A	Choline	-C(CH_3_)=CH_2_	CH_3_	2,4-D			
	A	Choline	-C(CH_3_)=CH_2_	CH_3_	Dicamba			
3	A	MDEA	Tallow ^5^	CH_3_	MCPA			[12]
	A	MDEA	Tallow	CH_3_	MCPP	Herbicidal		
	A	MDEA	Tallow	CH_3_	2,4-D			
4	B	Choline	2,4-D	C_6_H_13_	Br	Herbicidal	Esterified form of herbicide exhibits higher activity and is non-volatile due to presence of ions	[26]
	B	Choline	2,4-D	C_10_H_21_	Br			
5	B	Choline	MCPA	C_6_H_13_	Br	Herbicidal		[26]
	B	Choline	MCPA	C_10_H_21_	Br			
6	B	Choline	MCPA	C_10_H_21_	Br	Herbicidal	Synthesized as precursors to esterquats with two active ingredients within the molecule	[9]
7	B	Choline	Dicamba	C_10_H_21_	Br	Herbicidal		[47]
	B	Choline	Dicamba	C_12_H_25_	Br			
	B	Choline	Dicamba	C_14_H_29_	Br			
8	C	Choline	2,4-D	C_10_H_21_	MCPA	Herbicidal	Introduction of additional active ingredient into the molecule, which contributes to enhancement of efficacy, broadening spectrum of activity and minimization of resistance issue	[9]
	C	Choline	2,4-D	C_10_H_21_	MCPP			
	C	Choline	2,4-D	C_10_H_21_	4-CPA ^6^			
	C	Choline	MCPA	C_10_H_21_	2,4-D			
	C	Choline	MCPA	C_10_H_21_	MCPP			
	C	Choline	MCPA	C_10_H_21_	4-CPA			
	C	Choline	MCPP	C_10_H_21_	2,4-D			
	C	Choline	MCPP	C_10_H_21_	2,4-D			
	C	Choline	MCPP	C_10_H_21_	MCPA			
	C	Choline	4-CPA	C_10_H_21_	MCPP			
	C	Choline	MCPA	C_10_H_21_	Clopyralid ^7^			
	C	Choline	MCPA	C_10_H_21_	Dicamba			
	C	Choline	4-CPA	C_10_H_21_	Clopyralid			
	C	Choline	4-CPA	C_10_H_21_	Dicamba			
9	C	Choline	MCPA	C_8_H_17_	MCPP	Herbicidal	Introduction of additional active ingredient into the molecule, which contributes to enhancement of efficacy, broadening spectrum of activity and minimization of resistance issue	[48]
	C	Choline	MCPA	C_9_H_19_	MCPP			
	C	Choline	MCPA	C_10_H_21_	MCPP			
	C	Choline	MCPA	C_11_H_23_	MCPP			
	C	Choline	MCPA	C_12_H_25_	MCPP			
	C	Choline	MCPA	C_14_H_29_	MCPP			
10	C	Choline	Dicamba	C_10_H_21_	4-CPA	Herbicidal	Introduction of additional active ingredient into the molecule, which contributes to enhancement of efficacy, broadening spectrum of activity and minimization of resistance issue	[47]
	C	Choline	Dicamba	C_12_H_25_	4-CPA			
	C	Choline	Dicamba	C_14_H_29_	4-CPA			
	C	Choline	Dicamba	C_10_H_21_	2,4-D			
	C	Choline	Dicamba	C_12_H_25_	2,4-D			
	C	Choline	Dicamba	C_14_H_29_	2,4-D			
	C	Choline	Dicamba	C_10_H_21_	MCPA			
	C	Choline	Dicamba	C_12_H_25_	MCPA			
	C	Choline	Dicamba	C_14_H_29_	MCPA			
	C	Choline	Dicamba	C_10_H_21_	MCPP			
	C	Choline	Dicamba	C_12_H_25_	MCPP			
	C	Choline	Dicamba	C_14_H_29_	MCPP			
	C	Choline	Dicamba	C_10_H_21_	Clopyralid			
	C	Choline	Dicamba	C_12_H_25_	Clopyralid			
	C	Choline	Dicamba	C_14_H_29_	Clopyralid			

^1^ MCPA: 2-methyl-4-chlorophenoxyacetate, ^2^ MCPP: (*R*)-2-(4-chloro-o-tolyloxy)propionate, ^3^ 2,4-D: 2,4-dichlorophenoxyacetate, ^4^ Dicamba: 3,6-dichloro-2-methoxybenzoate, ^5^ Tallow means the mixture of alkyl chains with length distribution as follows: 1% C_12_H_25_, 4% C_14_H_29_, 31% C_16_H_33_, 64% C_18_H_37_, ^6^ 4-CPA: 4-chlorophenoxyacetate, ^7^ Clopyralid: 3,6-dichloro-2-pyridinecarboxylate.

**Table 3 ijms-25-05761-t003:** Reported choline-type esterquats with other biologically active moieties in their structure. Detailed structures of all cations are presented in Appendix A.

Core	Moiety	n	R_1_ (Figure 6)	Anion	Activity	Available Metrics	Notes	Ref.
	Lidocaine-derived choline							
A	Lidocaine-derived choline	1	C_5_H_11_	Cl	Analgesic and surface activity	ND ^1^	Self-assembled JPs ^6^	[84]
A	Lidocaine-derived choline	1	C_7_H_15_	Cl		REC_50_ ^2^ 3.5 mmol/L CMC ^3^ 2.51 mM		
A	Lidocaine-derived choline	1	C_9_H_19_	Cl		-		
A	Lidocaine-derived choline	1	C_12_H_25_	Cl		-		
A	Lidocaine-derived choline	1	C_7_H_15_	Cl		-		
B	Lidocaine-derived choline	-	C_8_H_17_	Cl	-	EC_50_ (DPPH) ^4,5^ 3.8 mmol/L CMC 1.27 mM		
	**MDEA derivative**							
C	MDEA esterquat for delivery of cytostatic drugs	-	-	-	-	-	Delivery of cytostatic drugs and siRNA	[42]
	**Sinapine derivative**	**X**	**Y**	**Anion**				
D	Sinapine	OCH_3_	OCH_3_	Cl	Antimicrobial and antioxidant/Food preservative	EC_50_ (*E. coli*) ^4^ 0.0056 µg/mL EC_50_ (DPPH) ^4.5^ 18.1 nmol/L	Antioxidant antimicrobial, anti-UV, anti-inflammatory, anticancer properties	[57]
D	Coumaric acid	H	H	Cl		EC_50_ (*E. coli*) ^4^ > 0.0375 µg/mL EC_50_ (DPPH) ^4,5^ > 150 nmol/L		
D	Caffeic acid	OH	H	Cl		EC_50_ (*E. coli*) ^4^ 0.0017 µg/mL EC_50_ (DPPH) ^4,5^ 6.26 nmol/L		
D	Ferulic acid	OCH_3_	H	Cl		EC_50_ (*E. coli*) ^4^ 0.0103 µg/mL EC_50_ (DPPH) ^4,5^ 36.73 nmol/L		

^1^ ND—not determined; ^2^ REC_50_: relative effective concentration determined using rat sciatic nerve block test; ^3^ CMC: critical micelle concentration; ^4^ EC_50_ (*E. coli*): half maximal effective concentration against *Escherichia coli* (*E. coli*) bacteria; ^5^ EC_50_ (DPPH): Antioxidant activity determined by diphenyl-1-picrylhydrazyl (DPPH) radical scavenging; ^6^ Janus particles.

**Table 4 ijms-25-05761-t004:** The general methods of the synthetic methods for betaine-based esterquats. Structures of esterquats are presented in Appendix A.

Synthetic Procedures	Location	References
Synthesis from haloacyl halides	Appendix A	[3,16,17,18,98,99,100,101,102,103]
Synthesis from haloesters	Appendix A	[104,105,106,107,108,109,110,111,112]
Synthesis from chlorobetainyl chloride	Appendix A	[113,114,115]
Synthesis of glycine betaine by classic esterification	Appendix A	[10,116,117,118,119,120,121]
Synthesis by O-alkylation of betaines	Appendix A	[11,88]
Synthesis by anion exchange	Appendix A	[11,14]

**Table 5 ijms-25-05761-t005:** Reported betaine-type esterquats with biologically active moiety in their structure.

No	Cation	Anion	Moiety Introduced	Activity/Potential Application	Available Metrics	Notes	Ref.
1	R_1_ = from C_2_H_5_ to C_18_H_37_	IS-M ^1^	IS-M	herbicidal/agrochemical	Application dose–10 g a.i./ha ^2^	All compounds were as effective as commercial product without addition of adjuvant	[11]
2	R_1_ = from C_2_H_5_ to C_18_H_37_	Dicamba ^3^	Dicamba	herbicidal/agrochemical	Application dose–200 g a.i./ha	All compounds were more effective than commercial active ingredient (sodium salt of dicamba) product without addition of adjuvant	[14]

^1^ IS-M—Iodosulfuron methyl; ^2^ a.i./ha—active ingredient applied per hectare; ^3^ Dicamba—3,6-dichloro-2-methoxybenzoic acid.

**Table 6 ijms-25-05761-t006:** Overview of the hydrolysis rate of esterquats (Note, that the rates of hydrolysis are not always linear).

Entry	Cation				Anion	Hydrolysis Rate Time to Complete Hydrolysis	Conditions	Ref.
	Core	R_1_	n	Y				
**Choline-type esterquats**								
1	**A**	Tallow ^1^	-	-	CH_3_SO_4_^−^	0.5% per month	Atmospheric	[65]
2	**B**	C_8_H_17_	1	-	Cl	Complete/200 min *	21 °C, pH = 4	[77]
3	**B**	C_10_H_21_	1	-	Cl	>300 min *	21 °C, pH = 4	
4	**B**	C_10_H_21_	1	-	Cl	160 days *	21 °C, pH = 8	
5	**B**	C_10_H_21_	1	-	Cl	Stable ^2^, >>365 days *	21 °C, pH = 10	
6	**B**	C_12_H_25_	1	-	Cl	>19 h *	21 °C, pH = 3	
7	**B**	C_12_H_25_	1	-	Cl	>22.5 h *	21 °C, pH = 4	
8	**B**	C_12_H_25_	1	-	Cl	120 min *	50 °C, pH = 4	
9	**B**	C_14_H_29_	1	-	Cl	270 min *	50 °C, pH = 4	
10	**B**	C_16_H_33_	1	-	Cl	110 min *	50 °C, pH = 4	
11	**C**	C_10_H_21_	1	-	2Cl	43% in 20 h	20 °C, 5 eq. NaOH	[78]
12	**C**	C_10_H_21_	1	-	2Cl	62% in 20 h	20 °C, 10 eq. NaOH	
13	**C**	C_14_H_29_	1	-	2Cl	10% in 20 h	80 °C, 75% (*v*/*v*) AcOH	
14	**C**	C_14_H_29_	1	-	2Cl	12% in 20 h	80 °C, 10 eq. HCl	
15	**C**	C_14_H_29_	1	-	2Cl	92% in 20 h	80 °C, 10 eq. NaOH	
16	**C**	C_14_H_29_	1	-	2Cl	2% in 20 h	80 °C, water	
17	**D**	C_12_H_25_	1	-	2I	9% in 10 days	25 °C, pH = 7 (non-buffered water)	[81]
18	**D**	C_12_H_25_	1	-	2I	50% in 6 days	25 °C, pH = 7 (phosphate-buffered water)	
19	**D**	C_12_H_25_	1	-	2I	Unstable ** ^3^	40 °C, pH = 7 (non-buffered water)	
20	**D**	C_12_H_25_	2	-	2I	1% in 10 days	25 °C, pH = 7 (non-buffered water)	
21	**D**	C_12_H_25_	2	-	2I	5% in 10 days	25 °C, pH = 7 (phosphate-buffered water)	
22	**D**	C_12_H_25_	2	H	2I	40% in 8 days	40 °C, pH = 7 (phosphate-buffered water)	
23	**E**	C_12_H_25_	2	H	2I	2% in 20 days	25 °C, pH = 7 (non-buffered water)	[82]
24	**E**	C_12_H_25_	2	H	2I	2% in 20 days	25 °C, pH = 4 (acetate-buffered water)	
25	**E**	C_12_H_25_	2	H	2I	4% in 20 days	25 °C, pH = 7 (phosphate-buffered water)	
26	**E**	C_12_H_25_	1	CH_3_	2I	3% in 20 days	25 °C, pH = 7 (non-buffered water)	
27	**E**	C_12_H_25_	1	CH_3_	2I	10% in 20 days	25 °C, pH = 4 (acetate-buffered water)	
28	**E**	C_12_H_25_	1	CH_3_	2I	20% in 20 days	25 °C, pH = 7 (phosphate-buffered water)	
29	**F**	C_14_H_29_	1	-	Br	Stable ** ^2^	0.4 M HCl	[47]
30	**F**	C_14_H_29_	1	-	Br	30 min	0.4 M NaOH	
31	**F**	C_14_H_29_	1	-	Dicamba ^4^	Stable ** ^2^	0.4 M HCl	
32	**F**	C_14_H_29_	1	-	Dicamba	30 min	0.4 M NaOH	
**Betaine-type esterquats**								
33	**G**	C_14_H_29_	1	-	Cl	Stable ** ^2^	30 °C, pH = 3	[16]
34	**G**	C_14_H_29_	1	-	Cl	90% in 24 h	30 °C, pH = 5	
35	**G**	C_14_H_29_	1	-	Cl	15% in 18 h	30 °C, pH = 6	
36	**G**	C_14_H_29_	1	-	Cl	100% in 20 h	25 °C, pH = 7	
37	**G**	C_14_H_29_	1	-	Cl	90% in 4 h	30 °C, pH = 8	
38	**G**	C_14_H_29_	1	-	Cl	90% in 2 h	30 °C, pH = 9	
39	**G**	C_14_H_29_	1	-	Cl	100% in 5 h	25 °C, pH = 7.9 (phosphate buffer)	
40	**G**	C_14_H_29_	1	-	Cl	100% in 5 h	25 °C, pH = 7.9 (phosphate buffer), 0.1 M NaCl	
41	**G**	C_14_H_29_	1	-	Cl	25% in 5 h	25 °C, pH = 7.9 (phosphate buffer), 0.5 M NaCl	
42	**G**	C_14_H_29_	1	-	Cl	15% in 5 h	25 °C, pH = 7.9 (phosphate buffer), 1 M NaCl	
43	**H**	C_8_H_17_	1	(CH_2_)_2_	2Cl	1 min	0.025 M H_2_SO_4_	[18]
44	**H**	C_8_H_17_	1	(CH_2_)_2_	2Cl	10 s	0.025 M NaOH	
45	**H**	C_10_H_21_	1	(CH_2_)_2_	2Cl	34 min	0.025 M H_2_SO_4_	
46	**H**	C_10_H_21_	1	(CH_2_)_2_	2Cl	15 s	0.025 M NaOH	
47	**H**	C_12_H_25_	1	(CH_2_)_2_	2Cl	210 min	0.025 M H_2_SO_4_	
48	**H**	C_12_H_25_	1	(CH_2_)_2_	2Cl	20 s	0.025 M NaOH	
49	**H**	C_8_H_17_	1	(CH_2_)_3_	2Cl	1 min	0.025 M H_2_SO_4_	
50	**H**	C_8_H_17_	1	(CH_2_)_3_	2Cl	10 s	0.025 M NaOH	
51	**H**	C_10_H_21_	1	(CH_2_)_3_	2Cl	41 min	0.025 M H_2_SO_4_	
52	**H**	C_10_H_21_	1	(CH_2_)_3_	2Cl	10 s	0.025 M NaOH	
53	**H**	C_12_H_25_	1	(CH_2_)_3_	2Cl	255 min	0.025 M H_2_SO_4_	
54	**H**	C_12_H_25_	1	(CH_2_)_3_	2Cl	20 s	0.025 M NaOH	
55	**H**	C_8_H_17_	1	(CH_2_)_4_	2Cl	1 min	0.025 M H_2_SO_4_	
56	**H**	C_8_H_17_	1	(CH_2_)_4_	2Cl	10 s	0.025 M NaOH	
57	**H**	C_10_H_21_	1	(CH_2_)_4_	2Cl	46 min	0.025 M H_2_SO_4_	
58	**H**	C_10_H_21_	1	(CH_2_)_4_	2Cl	15 s	0.025 M NaOH	
59	**H**	C_12_H_25_	1	(CH_2_)_4_	2Cl	340 min	0.025 M H_2_SO_4_	
60	**H**	C_12_H_25_	1	(CH_2_)_4_	2Cl	25 s	0.025 M NaOH	
61	**I**	C_10_H_21_	1	-	2Br	Unstable ** ^3^	pH = 4	[107]
62	**I**	C_10_H_21_	1	-	2Br	16 h	pH = 7	
63	**I**	C_10_H_21_	1	-	2Br	Unstable ** ^3^	pH = 10	
64	**J**	C_10_H_21_	1	-	2Br	Unstable ** ^3^	pH = 4	[107]
65	**J**	C_10_H_21_	1	-	2Br	6 h	pH = 7	
66	**J**	C_10_H_21_	1	-	2Br	Unstable ** ^3^	pH = 10	
67	**K**	C_18_H_37_	1	-	CH_3_SO_4_	Stable ** ^2^	RT, pH = 3.5	[116]
68	**K**	C_18_H_37_	1	-	CH_3_SO_4_	Stable ** ^2^	RT, pH = 4.5	
69	**K**	C_18_H_37_	1	-	CH_3_SO_4_	Stable ** ^2^	RT, pH = 5.6	
70	**K**	C_18_H_37_	1	-	CH_3_SO_4_	80% in 20 days	RT, pH = 6.6	
71	**K**	C_18_H_37_	1	-	CH_3_SO_4_	90% in 20 days	RT, pH = 7.5	
72	**K**	C_18_H_37_	1	-	CH_3_SO_4_	100% in 15 days	RT, pH = 8.1	

*—data refers to orthoesters that do not contain carbonyl groups; therefore, their susceptibility to hydrolysis at different pH values is compared to choline type esterquats; **—data not specified, ^1^ Tallow fatty acids, chain length distribution not provided; ^2^ Stable: no major changes during the measurement time; ^3^ Unstable-instant decomposition; ^4^ Dicamba: 3,6-dichloro-2-methoxybenzoate.

**Table 7 ijms-25-05761-t007:** Overview of biodegradation studies on esterquats.

Entry	Core	Structural Features of the Cation	Anion	Test	Biodegradation [%]	Ref.
**Choline-type esterquats**						
1	**A**	n = 1, R_1_ = C_12_H_25_, R_2_ = C_2_H_4_	2Cl	OECD 301C	58 *	[69]
2	**A**	n = 1, R_1_ = C_9_H_19_, R_2_ = CH_2_	2Br	OECD 301D	45 **	[19]
3	**A**	n = 1, R_1_ = C_11_H_23_, R_2_ = CH_2_	2Br	OECD 301D	50 **	[3]
4	**A**	n = 1, R_1_ = C_9_H_19_, R_2_ = C_4_H_8_	2Br	OECD 301D	40 **	[19]
5	**A**	n = 1, R_1_ = C_9_H_19,_ R_2_ = CH_2_OH	2Cl	OECD 301C	58 *	[61]
6	**A**	n = 1, R_1_ = C_11_H_23,_ R_2_ = CH_2_OH	2Cl	OECD 301C	58 *	
7	**A**	n = 1, R_1_ = C_13_H_27,_ R_2_ = CH_2_OH	2Cl	OECD 301C	58 *	
8	**A**	n = 1, R_1_ = C_12_H_25_, R_2_ = *trans*-CH=CH	2Cl	OECD 301C	52 *	[69]
9	**A**	n = 1, R_1_ = C_12_H_25_, R_2_ = *cis*-CH=CH	2Cl	OECD 301C	55 *	
10	**A**	n = 2, R_1_ = C_12_H_25,_ R_2_ = CH_2_OH	2Cl	OECD 301C	59 *	
11	**A**	n = 3, R_1_ = C_9_H_19,_ R_2_ = CH_2_OH	2Cl	OECD 301C	50 *	[61]
12	**A**	n = 5, R_1_ = C_8_H_17_, R_2_ = *trans*-CH=CH	2Cl	OECD 301C	52 *	[69]
13	**A**	n = 5, R_1_ = C_7_H_15,_ R_2_ = CH_2_OH	2Cl	OECD 301C	40 *	[61]
14	**B**	R_1_ = C_8_H_17_	MCPP ^1^	OECD 301F	12 **	[48]
15	**B**	R_1_ = C_9_H_19_	MCPP	OECD 301F	11 **	
16	**B**	R_1_ = C_10_H_21_	MCPP	OECD 301F	9 **	
17	**B**	R_1_ = C_11_H_23_	MCPP	OECD 301F	6 **	
18	**B**	R_1_ = C_12_H_25_	MCPP	OECD 301F	4 **	
19	**B**	R_1_ = C_14_H_29_	MCPP	OECD 301F	2 **	
20	**C**	n = 1, R_1_ = CHCH_2_	Cl	OECD 301F	73 **	[12]
21	**C**	n = 1, R_1_ = C(CH_3_)CH_2_	Cl	OECD 301F	70 **	
22	**C**	n = 1, R_1_ = C_11_H_23_	Cl	OECD 301C	78 *	[61]
23	**C**	n = 1, R_1_ = C_9_H_19_	Cl	OECD 301D	90 **	[19]
24	**C**	n = 1, R_1_ = CHCH_2_	MCPA ^2^	OECD 301F	37 **	[12]
25	**C**	n = 1, R_1_ = C(CH_3_)CH_2_	MCPA	OECD 301F	29 **	
26	**D**	R_1_ = tallow	Cl	OECD 301F	91 **	[12]
27	**D**	R_1_ = tallow	MCPA	OECD 301F	63 **	
28	**E**	n = 1, Y = H, R_1_ = C_11_H_23_, R_2_ = CH_3_	Br	OECD 301D	95 **	[3]
29	**F**	n = 1, R_1_ = C_12_H_25_	2Cl	OECD 301C	52 *	[69]
30	**G**	n = 1, R_1_ = C_10_H_21_	2I	OECD 301C	60 **	[81]
31	**G**	n = 1, R_1_ = C_12_H_25_	2I	OECD 301C	70 **	
32	**G**	n = 1, R_1_ = C_14_H_29_	2I	OECD 301C	60 **	
33	**G**	n = 2, R_1_ = C_10_H_21_	2I	OECD 301C	30 **	
34	**G**	n = 2, R_1_ = C_12_H_25_	2I	OECD 301C	45 **	
35	**G**	n = 2, R_1_ = C_14_H_29_	2I	OECD 301C	20 **	
36	**H**	n = 1, Y = CH_3,_ R_1_ = C_12_H_25_ R_2_ = CH_2_	2I	OECD 301C	31 **	[82]
37	**H**	n = 2, Y = H_,_ R_1_ = C_12_H_25_, R_2_ = CH_2_	2I	OECD 301C	25 **	
38	**I**	n = 1, Y = H_,_ R_1_ = C_12_H_25_	I	OECD 301C	60 **	[82]
39	**I**	n = 1, Y = CH_3,_ R_1_ = C_12_H_25_	I	OECD 301C	35 **	
40	**J**	n = 2, m = 1, R_1_ = C_12_H_25_	2I	OECD 301C	60 **	[82]
41	**J**	n = 2, m = 2, R_1_ = C_12_H_25_	2I	OECD 301C	29 **	
**Betaine-type esterquats**						
42	**K**	n = 1, R_1_ = C_12_H_25_	Br	OECD 301D	85 **	[3]
43	**L**	n = 1, R_1_ = C_12_H_25_	2Br	OECD 301D	55 **	
44	**M**	m = 7	2Cl	OECD 310	31 **	[111]
45	**M**	m = 9	2Cl	OECD 310	39 **	
46	**N**	m = 7	3Cl	OECD 310	41 **	
47	**N**	m = 9	3Cl	OECD 310	42 **	
48	**N**	m = 11	3Cl	OECD 310	52 **	

* for 14 days; ** for 28 days; ^1^ MCPP: (R)-2-(4-chloro-o-tolyloxy)propionate; ^2^ MCPA: 2-methyl-4-chlorophenoxyacetate.

**Table 8 ijms-25-05761-t008:** Overview of reported results of antimicrobial activity of esterquats.

No	Core; Cation	Anion	Parameter (Units)	Antimicrobial Activity	Ref.
1	**A**; R_1_ = C_11_H_23_, R_2_ = C_8_H_17_	Br	IZD ^1^ (mm) (at 1000 µg/mL)	*C. tropicalis*: 15; *E. coli*: 10; *S. aureus*: 9	[67]
2	**A**; R_1_ = C_13_H_27_, R_2_ = C_8_H_17_	Br	IZD (mm) (at 1000 µg/mL)	*C. tropicalis*: 13; *E. coli*: 11; *S. aureus*: 12	
3	**A**; R_1_ = C_15_H_31_, R_2_ = C_8_H_17_	Br	IZD (mm) (at 1000 µg/mL)	*C. tropicalis*: 13; *E. coli*: 13; *S. aureus*: 15	
4	**B**; n = 1, Y = H, R_1_ = C_11_H_23_, R_2_ = C_5_H_11_	2Cl	MIC ^2^ (µg/mL)	*B. subtilis*: 256; *E. coli*: >512; *R. rubra*: >512; *S. aureus*: >512; *S. marcescens*: >512	[73]
5	**B**; n = 1, Y = H, R_1_ = C_11_H_23_, R_2_ = C_6_H_13_	2Cl	MIC (µg/mL)	*B. subtilis*: 256; *E. coli*: >512; *R. rubra*: >512; *S. aureus*: >512; *S. marcescens*: >512	
6	**B**; n = 1, Y = H, R_1_ = C_11_H_23_, R_2_ = C_8_H_17_	2Cl	MIC (µg/mL)	*B. subtilis*: 256; *E. coli*: >512; *R. rubra*: >512; *S. aureus*: >512; *S. marcescens*: >512	
7	**B**; n = 2, Y = H, R_1_ = C_9_H_19_, R_2_ = C_4_H_9_	2Cl	IZD (mm) (at 1000 µg/mL)	*A. Fumigatus*: 18.6; *B. subtilis*: 21.3; *E. coli*: 15.9; *S. pneumonia*: 19.6	[74]
8	**B**; n = 2, Y = H, R_1_ = C_9_H_19_, R_2_ = C_5_H_11_	2Cl	IZD (mm) (at 1000 µg/mL)	*A. Fumigatus*: 15.7; *B. subtilis*: 19.8; *E. coli*: 15.3; *S. pneumonia*: 16.2	
9	**B**; n = 2, Y = H, R_1_ = C_9_H_19_, R_2_ = C_6_H_13_	2Cl	MIC (µg/mL)	*B. subtilis*: 256; *E. coli*: >512; *R. rubra*: >512; *S. aureus*: 512; *S. marcescens*: >512	[73]
10	**B**; n = 2, Y = H, R_1_ = C_9_H_19_, R_2_ = C_6_H_13_	2Cl	IZD (mm) (at 1000 µg/mL)	*A. Fumigatus*: 12.9; *B. subtilis*: 15.9; *E. coli*: 12.9; *S. pneumonia*: 15.1	[74]
11	**B**; n = 2, Y = CH_3_, R_1_ = C_9_H_19_, R_2_ = C_6_H_13_	2Cl	MIC (µg/mL)	*B. subtilis*: 512; *E. coli*: >512; *R. rubra*: >512; *S. aureus*: 256; *S. marcescens*: >512	[73]
12	**B**; n = 2, Y = H, R_1_ = C_9_H_19_, R_2_ = C_8_H_17_	2Cl	MIC (µg/mL)	*B. subtilis*: >256; *E. coli*: >512; *R. rubra*: >512; *S. aureus*: >512; *S. marcescens*: >512	
13	**B**; n = 2, Y = CH_3_, R_1_ = C_9_H_19_, R_2_ = C_8_H_17_	2Cl	MIC (µg/mL)	*B. subtilis*: 256; *E. coli*: >512; *R. rubra*: >512; *S. aureus*: 256; *S. marcescens*: >512	
14	**B**; n = 2, Y = H, R_1_ = C_11_H_23_, R_2_ = C_4_H_9_	2Cl	IZD (mm) (at 1000 µg/mL)	*A. Fumigatus*: 17. 8; *B. subtilis*: 22. 4; *E. coli*: 16.9; *S. pneumonia*: 20.6	[74]
15	**B**; n = 2, Y = H, R_1_ = C_11_H_23_, R_2_ = C_5_H_11_	2Cl	MIC (µg/mL)	*B. subtilis*: 256; *E. coli*: >512; *R. rubra*: >512; *S. aureus*: 512; *S. marcescens*: >512	[73]
16	**B**; n = 2, Y = H, R_1_ = C_11_H_23_, R_2_ = C_6_H_13_	2Cl	MIC (µg/mL)	*B. subtilis*: 128; *E. coli*: >512; *R. rubra*: >512; *S. aureus*: >512; *S. marcescens*: >512	
17	**B**; n = 2, Y = H, R_1_ = C_11_H_23_, R_2_ = C_5_H_11_	2Cl	IZD (mm) (at 1000 µg/mL)	*A. Fumigatus*: 17.6; *B. subtilis*: 19.9; *E. coli*: 16.9; *S. pneumonia*: 18.4	[74]
18	**B**; n = 2, Y = CH_3_, R_1_ = C_11_H_23_, R_2_ = C_5_H_11_	2Cl	MIC (µg/mL)	*B. subtilis*: 256; *E. coli*: >512; *R. rubra*: >512; *S. aureus*: 256; *S. marcescens*: >512	[73]
19	**B**; n = 2, Y = H, R_1_ = C_11_H_23_, R_2_ = C_6_H_13_	2Cl	IZD (mm) (at 1000 µg/mL)	*A. Fumigatus*: 16.3; *B. subtilis*: 18.3; *E. coli*: 15.9; *S. pneumonia*: 17.3	[74]
20	**B**; n = 2, Y = CH_3_, R_1_ = C_11_H_23_, R_2_ = C_6_H_13_	2Cl	MIC (µg/mL)	*B. subtilis*: 256; *E. coli*: >512; *R. rubra*: >512; *S. aureus*: 256; *S. marcescens*: >512	[73]
21	**B**; n = 2, Y = H, R_1_ = C_11_H_23_, R_2_ = C_8_H_17_	2Cl	MIC (µg/mL)	*A. Fumigatus*: 3.9; *B. subtilis*: 0.49; *E. coli*: 15.63; *S. pneumonia*: 0.98	[74]
			IZD (mm) (at 1000 µg/mL)	*A. Fumigatus*: 12.6; *B. subtilis*: 15.4; *E. coli*: 12.7; *S. pneumonia*: 14.6	
22	**B**; n = 2, Y = H, R_1_ = C_11_H_23_, R_2_ = C_8_H_17_	2Cl	MIC (µg/mL)	*B. subtilis*: 256; *E. coli*: >512; *R. rubra*: >512; *S. aureus*: 512; *S. marcescens*: >512;	[73]
23	**B**; n = 2, Y = CH_3_, R_1_ = C_11_H_23_, R_2_ = C_8_H_17_	2Cl	MIC (µg/mL)	*B. subtilis*: 256; *E. coli*: >512; *R. rubra*: >512; *S. aureus*: 256; *S. marcescens*: >512	
24	**B**; n = 2, Y = CH_3_, R_1_ = C_13_H_27_, R_2_ = C_6_H_13_	2Cl	MIC (µg/mL)	*B. subtilis*: 512; *E. coli*: >512; *R. rubra*: >512; *S. aureus*: >512; *S. marcescens*: >512	
25	**B**; n = 2, Y = H, R_1_ = C_13_H_27_, R_2_ = C_8_H_17_	2Cl	MIC (µg/mL)	*B. subtilis*: >512; *E. coli*: >512; *R. rubra*: >512; *S. aureus*: >512; *S. marcescens*: >512	
26	**B**; n = 2, Y = CH_3_, R_1_ = C_13_H_27_, R_2_ = C_8_H_17_	2Cl	MIC (µg/mL)	*B. subtilis*: >512; *E. coli*: >512; *R. rubra*: >512; *S. aureus*: >512; *S. marcescens*: >512	
27	**C**; n = 1, R_1_ = C_12_H_25_	2I	MIC (µg/mL)	*A. niger*: 400; *B. subtilis*: 10; *C. albicans*: 400; *E. coli*: 25; *M. gypseum*: 25; *M. luteus*: 50; *P. aeruginosa*: 400; *P. chrysogenum*: 400; *S. aureus*: 25; *S. cerevisiae*: 400; *S. typhimurium*: 100; *T. mentagrophytes*: 50	[81]
28	**C**; n = 2, R_1_ = C_12_H_25_	2I	MIC (µg/mL)	*A. niger*: >400; *B. subtilis*: 5; *C. albicans*: 400; *E. coli*: 10; *M. gypseum*: 400; *M. luteus*: 5; *P. aeruginosa*: 200; *P. chrysogenum*: >400; S. aureus: 5; *S. cerevisiae*: >400; *S. typhimurium*: 100; *T. mentagrophytes*: 200	
29	**D**; n = 2, Y = H, R_1_ = C_12_H_25_, R_2_ = CH_2_	2I	MIC (µg/mL)	*A. niger*: >400; *B. subtilis*: 200; *C. albicans*: >400; *E. coli*: 400; *M. gypseum*: 400; *M. luteus*: 400; *P. aeruginosa*: 400; *P. chrysogenum*: >400; *S. aureus*: 200; *S. cerevisiae*: >400; *S. typhimurium*: >400; *T. mentagrophytes*: >400	[82]
30	**D**; n = 1, Y = CH_3_, R_1_ = C_12_H_25_, R_2_ = CH_2_	I	MIC (µg/mL)	*A. niger*: >400; *B. subtilis*: 100; *C. albicans*: >400; *E. coli*: 200; *M. gypseum*: 400; *M. luteus*: 100; *P. aeruginosa*: >400; *P. chrysogenum*: >400; *S. aureus*: 100; *S. cerevisiae*: >400; *S. typhimurium*: >400; *T. mentagrophytes*: 200	
31	**E**; n = 1, Y = H, R_1_ = C_12_H_25_	I	MIC (µg/mL)	*A. niger*: >400; *B. subtilis*: 2.5; *C. albicans*: >400; *E. coli*: 10; *M. gypseum*: 10; *M. luteus*: 5; *P. aeruginosa*: 100; *P. chrysogenum*: 200; *S. aureus*: 2.5; *S. cerevisiae*: 400; *S. typhimurium*: 200; *T. mentagrophytes*: 50	
32	**E**; n = 1, Y = CH_3_, R_1_ = C_12_H_25_	I	MIC (µg/mL)	*A. niger*: >400; *B. subtilis*: 5; *C. albicans*: 400; *E. coli*: >400; *M. gypseum*: 5; *M. luteus*: 2.5; *P. aeruginosa*: >400; *P. chrysogenum*: 100; *S. aureus*: 2.5; *S. cerevisiae*: 400; *S. typhimurium*: >400; *T. mentagrophytes*: 400	
33	**F**; n = 2, m = 1, R_1_ = C_12_H_25_	2I	MIC (µg/mL)	*A. niger*: >400; *B. subtilis*: 10; *C. albicans*: 400; *E. coli*: 200; *M. gypseum*: 25; *M. luteus*: 25; *P. aeruginosa*: 200; *P. chrysogenum*: 400; *S. aureus*: 5; *S. cerevisiae*: 400; *S. typhimurium*: 400; *T. mentagrophytes*: 100	
34	**F**; n = 2, m = 2, R_1_ = C_12_H_25_	2I	MIC (µg/mL)	*A. niger*: >200; *B. subtilis*: 50; *C. albicans*: >200; *E. coli*: 200; *M. gypseum*: 100; *M. luteus*: 100; *P. aeruginosa*: >200; *P. chrysogenum*: >200; *S. aureus*: 100; *S. cerevisiae*: >200; *S. typhimurium*: >200; *T. mentagrophytes*: >200	
35	**G**; R_2_ = C_10_H_21_	Br	MIC (µg/mL)	*E. coli*: 6.42; *P. putida*: 12.83	[47]
			MBC ^3^ (µg/mL)	*E. coli*: 12.83; *P. putida*: 25.67	
			EC_50_ ^4^ (µg/mL)	*E. coli*: 8.73; *P. putida*: 14.89	
36	**G**; R_1_ = C_12_H_25_	Br	MIC (µg/mL)	*E. coli*: 6.77; *P. putida*: 13.53	
			MBC (µg/mL)	*E. coli*: 54.14; *P. putida*: >54.14	
			EC_50_ (µg/mL)	*E. coli*: 10.83; *P. putida*: 16.78	
37	**G**; R_2_ = C_14_H_29_	Br	MIC (µg/mL)	*E. coli*: 14.24; *P. putida*: 284.72	
			MBC (µg/mL)	*E. coli*: 56.94; *P. putida*: >56.94	
			EC_50_ (µg/mL)	*E. coli*: 53.53; *P. putida*: 48.97	
38	**H**; R_2_ = C_8_H_17_	MCPP ^5^	MIC (µg/mL)	*B. subtilis*: 74.83; *C. albicans*: 149.67; *P. putida*: 299.33	[48]
			MBC (µg/mL)	*B. subtilis*: 299.33; *C. albicans*: 299.33; *P. putida*: 598.67	
39	**H**; R_2_ = C_9_H_19_	MCPP	MIC (µg/mL)	*B. subtilis*: 76.59; *C. albicans*: 153.18; *P. putida*: 306.35	
			MBC (µg/mL)	*B. subtilis*: 306.35; *C. albicans*: 306.35; *P. putida*: 612.7	
40	**H**; R_2_ = C_10_H_21_	MCPP	MIC (µg/mL)	*B. subtilis*: 78.34; *C. albicans*: 78.34; *P. putida*: 313.37	
			MBC (µg/mL)	*B. subtilis*: 313.37; *C. albicans*: 156.68; *P. putida*: 626.73	
41	**H**; R_2_ = C_11_H_23_	MCPP	MIC (µg/mL)	*B. subtilis*: 78.34; *C. albicans*: 78.34; *P. putida*: 78.34	
			MBC (µg/mL)	*B. subtilis*: 313.37; *C. albicans*: 156.68; *P. putida*: 626.73	
42	**H**; R_2_ = C_12_H_25_	MCPP	MIC (µg/mL)	*B. subtilis*: 20.49; *C. albicans*: 81.85; *P. putida*: 81.85	
			MBC (µg/mL)	*B. subtilis*: 163.7; *C. albicans*: 163.7; *P. putida*: 327.39	
43	**H**; R_2_ = C_14_H_29_	MCPP	MIC (µg/mL)	*B. subtilis*: 5.33; *C. albicans*: 8.54; *P. putida*: 34.14	
			MBC (µg/mL)	*B. subtilis*: 21.37; *C. albicans*: 17.07; *P. putida*: 85.36	
44	**I**; R_2_ = C_10_H_21_	4-CPA ^6^	MIC (µg/mL)	*E. coli*: 15.48; *P. putida*: 15.48	[47]
			MBC (µg/mL)	*E. coli*: 30.95; *P. putida*: 30.95	
			EC_50_ (µg/mL)	*E. coli*: 16.71; *P. putida*: 22.9	
45	**I**; R_2_ = C_12_H_25_	4-CPA	MIC (µg/mL)	*E. coli*: 8.09; *P. putida*: 16.18	
			MBC (µg/mL)	*E. coli*: 32.35; *P. putida*: 32.35	
			EC_50_ (µg/mL)	*E. coli*: 20.71; *P. putida*: 20.71	
46	**I**; R_2_ = C_14_H_29_	4-CPA	MIC (µg/mL)	*E. coli*: 33.76; *P. putida*: 33.76	
			MBC (µg/mL)	*E. coli*: 67.51; *P. putida*: >67.51	
			EC_50_ (µg/mL)	*E. coli*: 62.11; *P. putida*: 56.71	
47	**I**; R_2_ = C_10_H_21_	2,4-D ^7^	MIC (µg/mL)	*E. coli*: 16.34; *P. putida*: 32.67	
			MBC (µg/mL)	*E. coli*: 32.67; *P. putida*: 65.35	
			EC_50_ (µg/mL)	*E. coli*: 18.3; *P. putida*: 33.98	
48	**I**; R_2_ = C_12_H_25_	2,4-D	MIC (µg/mL)	*E. coli*: 8.52; *P. putida*: 17.04	
			MBC (µg/mL)	*E. coli*: 17.04; *P. putida*: 34.08	
			EC_50_ (µg/mL)	*E. coli*: 15.67; *P. putida*: 31.35	
49	**I**; R_2_ = C_14_H_29_	2,4-D	MIC (µg/mL)	*E. coli*: 35.48; *P. putida*: 35.48	
			MBC (µg/mL)	*E. coli*: 70.96; *P. putida*: >70.96	
			EC_50_ (µg/mL)	*E. coli*: 45.41; *P. putida*: 84.44	
50	**I**; R_2_ = C_10_H_21_	MCPA ^8^	MIC (µg/mL)	*E. coli*: 15.83; *P. putida*: 15.83	
			MBC (µg/mL)	*E. coli*: 31.65; *P. putida*: 31.65	
			EC_50_ (µg/mL)	*E. coli*: 13.29; *P. putida*: 21.52	
51	**I**; R_2_ = C_12_H_25_	MCPA	MIC (µg/mL)	*E. coli*: 8.26; *P. putida*: 16.53	
			MBC (µg/mL)	*E. coli*: 16.53; *P. putida*: 16.53	
			EC_50_ (µg/mL)	*E. coli*: 13.22; *P. putida*: 20.49	
52	**I**; R_2_ = C_14_H_29_	MCPA	MIC (µg/mL)	*E. coli*: 34.46; *P. putida*: 34.46	
			MBC (µg/mL)	*E. coli*: 68.92; *P. putida*: >68.92	
			EC_50_ (µg/mL)	*E. coli*: 60.65; *P. putida*: 78.56	
53	**I**; R_2_ = C_10_H_21_	MCPP	MIC (µg/mL)	*E. coli*: 16.18; *P. putida*: 16.18	
			MBC (µg/mL)	*E. coli*: 32.35; *P. putida*: 32.35	
			EC_50_ (µg/mL)	*E. coli*: 31.06; *P. putida*: 26.53	
54	**I**; R_2_ = C_12_H_25_	MCPP	MIC (µg/mL)	*E. coli*: 8.44; *P. putida*: 16.88	
			MBC (µg/mL)	*E. coli*: 16.88; *P. putida*: 33.76	
			EC_50_ (µg/mL)	*E. coli*: 14.18; *P. putida*: 16.2	
55	**I**; R_2_ = C_14_H_29_	MCPP	MIC (µg/mL)	*E. coli*: 17.58; *P. putida*: 35.16	
			MBC (µg/mL)	*E. coli*: 70.32; *P. putida*: >70.32	
			EC_50_ (µg/mL)	*E. coli*: 67.51; *P. putida*: 170.87	
56	**I**; R_2_ = C_10_H_21_	Clopyralid ^9^	MIC (µg/mL)	*E. coli*: 7.81; *P. putida*: 15.61	
			MBC (µg/mL)	*E. coli*: 31.22; *P. putida*: 31.22	
			EC_50_ (µg/mL)	*E. coli*: 21.85; *P. putida*: 23.1	
57	**I**; R_2_ = C_12_H_25_	Clopyralid	MIC (µg/mL)	*E. coli*: 8.16; *P. putida*: 16.31	
			MBC (µg/mL)	*E. coli*: 16.31; *P. putida*: 32.62	
			EC_50_ (µg/mL)	*E. coli*: 16.31; *P. putida*: 11.09	
58	**I**; R_2_ = C_14_H_29_	Clopyralid	MIC (µg/mL)	*E. coli*: 17.01; *P. putida*: 34.03	
			MBC (µg/mL)	*E. coli*: 68.05; *P. putida*: >68.05	
			EC_50_ (µg/mL)	*E. coli*: 41.51; *P. putida*: 115.69	
59	**J**; X = OCH_3_, Y = OCH_3_	Cl	EC_50_ (µg/mL)	*E. coli*: 0.0056	[57]
60	**J**; X = H, Y = H	Cl	EC_50_ (µg/mL)	*E. coli*: >0.0375	
61	**J**; X = OH, Y = H	Cl	EC_50_ (µg/mL)	*E. coli*: 0.0017	
62	**J**; X = OCH_3_, Y = H	Cl	EC_50_ (µg/mL)	*E. coli*: 0.0103	
**Betaine-type esterquats**					
63	**K**; n = 1, m = 2, R_1_ = C_8_H_17_	2Cl	MIC (µg/mL)	*E. coli*: 512; *S. aureus*: 512	[18]
64	**K**; n = 1, m = 2, R_1_ = C_10_H_21_	2Cl	MIC (µg/mL)	*E. coli*: 128; *S. aureus*: 64	
65	**K**; n = 1, m = 2, R_1_ = C_12_H_25_	2Cl	MIC (µg/mL)	*E. coli*: 256; *S. aureus*: 128	
66	**K**; n = 1, m = 3, R_1_ = C_8_H_17_	2Cl	MIC (µg/mL)	*E. coli*: 512; *S. aureus*: 512	
67	**K**; n = 1, m = 3, R_1_ = C_10_H_21_	2Cl	MIC (µg/mL)	*E. coli*: 256; *S. aureus*: 128	
68	**K**; n = 1, m = 3, R_1_ = C_12_H_25_	2Cl	MIC (µg/mL)	*E. coli*: 256; *S. aureus*: 256	
69	**K**; n = 1, m = 4, R_1_ = C_8_H_17_	2Cl	MIC (µg/mL)	*E. coli*: 256; *S. aureus*: 256	
70	**K**; n = 1, m = 4, R_1_ = C_10_H_21_	2Cl	MIC (µg/mL)	*E. coli*: 64; *S. aureus*: 32	
71	**K**; n = 1, m = 4, R_1_ = C_12_H_25_	2Cl	MIC (µg/mL)	*E. coli*: 128; *S. aureus*: 64	
72	**L**; n = 1, R_1_ = C_12_H_25_	2Cl	IZD (mm) (at 400 µg/mL)	*B. subtilis*: 16; *C. albicans*: 21; *E. coli*: 10; *S. aureus*: 19	[20]
73	**L**; n = 1, R_1_ = C_14_H_29_	2Cl	IZD (mm) (at 400 µg/mL)	*B. subtilis*: 14; *C. albicans*: 18; *E. coli*: 11; *S. aureus*: 16	
74	**L**; n = 1, R_1_ = C_16_H_33_	2Cl	IZD (mm) (at 400 µg/mL)	*B. subtilis*: 13; *C. albicans*: 20; *E. coli*: 11; *S. aureus*: 11	
75	**M**; n = 1; R_1_ = C_12_H_25_	Doc ^10^	EC_50_ (µg/mL)	*Aliivibrio fischeri*: 4.61 (5 min); 3.47 (15 min); 3.19 (30 min); *Raphidocelis subcapitata*: 0.58	[21]
76	**M**; n = 1; R_1_ = C_12_H_25_	SCN	EC_50_ (µg/mL)	*Aliivibrio fischeri*: 1.41 (5 min); 0.77 (15 min); 0.61 (30 min); *Raphidocelis subcapitata*: 0.27	
77	**M**; n = 1; R_1_ = C_14_H_29_	Doc	EC_50_ (µg/mL)	*Aliivibrio fischeri*: 3.19 (5 min); 1.39 (15 min); 1.37 (30 min); *Raphidocelis subcapitata*: 0.74	
78	**M**; n = 1; R_1_ = C_14_H_29_	SCN	EC_50_ (µg/mL)	*Aliivibrio fischeri*: 1.61 (5 min); 1.10 (15 min); 0.90 (30 min); *Raphidocelis subcapitata*: 0.54	
79	**N**; n = 1, R_1_ = C_9_H_19_	2Br	MIC (µg/mL)	*E. coli*: >161.52; *E. faecalis*: >161.52; *P. aeruginosa*: >161.52; *S. aureus* (ATCC33592): >161.52; *S. aureus* (USA300-0114): 161.52; *S. aureus*: >161.52	[107]
80	**N**; n = 1, R_1_ = C_10_H_21_	2Br	MIC (µg/mL)	*E. coli*: >168.52; *E. faecalis*: >168.52; *P. aeruginosa*: >168.52; *S. aureus* (ATCC33592): 168.52; *S. aureus* (USA300-0114): 84.26; *S. aureus*: 168.52	
81	**N**; n = 1, R_1_ = C_11_H_23_	2Br	MIC (µg/mL)	*E. coli*: 44.23; *E. faecalis*: 44.23; *P. aeruginosa*: 87.76; *S. aureus* (ATCC33592): 22.47; *S. aureus* (USA300-0114): 11.23; *S. aureus*: 22.47	
82	**N**; n = 1, R_1_ = C_12_H_25_	2Br	MIC (µg/mL)	*E. coli*: 11.68; *E. faecalis*: 11.68; *P. aeruginosa*: 23.36; *S. aureus* (ATCC33592): 5.84; *S. aureus* (USA300-0114): 2.92; *S. aureus*: 1.46	
83	**N**; n = 1, R_1_ = C_13_H_27_	2Br	MIC (µg/mL)	*E. coli*: 6.06; *E. faecalis*: 12.13; *P. aeruginosa*: 24.26; *S. aureus* (ATCC33592): 6.06; *S. aureus* (USA300-0114): 6.06; *S. aureus*: 3.03	
84	**N**; n = 1, R_1_ = C_15_H_31_	2Br	MIC (µg/mL)	*E. coli*: 51.29; *E. faecalis*: 26.05; *P. aeruginosa*: 101.76; *S. aureus* (ATCC33592): 26.05; *S. aureus* (USA300-0114): 13.03; *S. aureus*: 13.03	
85	**N**; n = 1, R_1_ = C_17_H_35_	2Br	MIC (µg/mL)	*E. coli*: 217.52; *E. faecalis*: 27.84; *P. aeruginosa*: >217.52; *S. aureus* (ATCC33592): 27.84; *S. aureus* (USA300-0114): 13.92; *S. aureus*: 13.92	
86	**O**; n = 1, m = 8, R_1_ = C_6_H_5_CH_2_	Br	MIC (µg/mL)	*E. coli*: >100.04; *E. faecalis*: >100.04; *P. aeruginosa*: >100.04; *S. aureus* (ATCC33592): >100.04; *S. aureus* (USA300-0114): >100.04; *S. aureus*: >100.04	
87	**O**; n = 1, m = 9, R_1_ = C_6_H_5_CH_2_	Br	MIC (µg/mL)	*E. coli*: >103.54; *E. faecalis*: >103.54; *P. aeruginosa*: >103.54; *S. aureus* (ATCC33592): >103.54; *S. aureus* (USA300-0114): >103.54; *S. aureus*: >103.54	
88	**O**; n = 1, m = 10, R_1_ = C_6_H_5_CH_2_	Br	MIC (µg/mL)	*E. coli*: 107.04; *E. faecalis*: 107.04; *P. aeruginosa*: >107.04; *S. aureus* (ATCC33592): 53.52; *S. aureus* (USA300-0114): 107.04; *S. aureus*: 53.52	
89	**O**; n = 1, m = 11, R_1_ = C_6_H_5_CH_2_	Br	MIC (µg/mL)	*E. coli*: 55.27; *E. faecalis*: 55.27; *P. aeruginosa*: >110.54; *S. aureus* (ATCC33592): 27.86; *S. aureus* (USA300-0114): 14.15; *S. aureus*: 14.15	
90	**O**; n = 1, m = 12, R_1_ = C_6_H_5_CH_2_	Br	MIC (µg/mL)	*E. coli*: 28.74; *E. faecalis*: 14.6; *P. aeruginosa*: >114.04; *S. aureus* (ATCC33592): 14.6; *S. aureus* (USA300-0114): 7.3; *S. aureus*: 14.6	
91	**O**; n = 1, m = 14, R_1_ = C_6_H_5_CH_2_	Br	MIC (µg/mL)	*E. coli*: 7.75; *E. faecalis*: 1.94; *P. aeruginosa*: 30.5; *S. aureus* (ATCC33592): 0.97; *S. aureus* (USA300-0114): 0.97; *S. aureus*: 0.97	
92	**O**; n = 1, m = 16, R_1_ = C_6_H_5_CH_2_	Br	MIC (µg/mL)	*E. coli*: 4.1; *E. faecalis*: 2.05; *P. aeruginosa*: >128.04; *S. aureus* (ATCC33592): 0.51; *S. aureus* (USA300-0114): 1.02; *S. aureus*: 0.51	
93	**O**; n = 2, m = 1, R_1_ = C_4_H_9_	Sac ^11^	EC_50_ (µg/mL)	*A. fischeri*: 248.48 (5 min); 103.51 (15 min); 68.06 (30 min); *R. subcapitata*: 4.71	[21]
94	**O**; n = 4, m = 1, R_1_ = C_4_H_9_	Sac	EC_50_ (µg/mL)	*A. fischeri*: 1904.43 (5 min); 628.04 (15 min); 401.14 (30 min); *R. subcapitata*: 12.83	
95	**P**; n = 1, m = 5	2Br	MIC (µg/mL)	*E. coli*: >175.29; *E. faecalis*: >175.29; *P. aeruginosa*: >175.29; *S. aureus* (ATCC33592): >175.29; *S. aureus* (USA300-0114): >175.29; *S. aureus*: >175.29	[107]
96	**P**; n = 1, m = 6	2Br	MIC (µg/mL)	*E. coli*: >182.29; *E. faecalis*: >182.29; *P. aeruginosa*: >182.29; *S. aureus* (ATCC33592): >182.29; *S. aureus* (USA300-0114): 182.29; *S. aureus*: >182.29	
97	**P**; n = 1, m = 7	2Br	MIC (µg/mL)	*E. coli*: 189.29; *E. faecalis*: 47.7; *P. aeruginosa*: >189.29; *S. aureus* (ATCC33592): 47.7; *S. aureus* (USA300-0114): 24.23; *S. aureus*: >189.29	
98	**P**; n = 1, m = 8	2Br	MIC (µg/mL)	*E. coli*: 12.56; *E. faecalis*: 6.28; *P. aeruginosa*: 98.14; *S. aureus* (ATCC33592): 3.14; *S. aureus* (USA300-0114): 3.14; *S. aureus*: 3.14	
99	**P**; n = 1, m = 10	2Br	MIC (µg/mL)	*E. coli*: 1.63; *E. faecalis*: 3.25; *P. aeruginosa*: 13.01; *S. aureus* (ATCC33592): 1.63; *S. aureus* (USA300-0114): 1.63; *S. aureus*: 1.63	
100	**P**; n = 1, m = 12	2Br	MIC (µg/mL)	*E. coli*: 13.91; *E. faecalis*: 27.81; *P. aeruginosa*: 27.81; *S. aureus* (ATCC33592): 6.95; *S. aureus* (USA300-0114): 6.95; *S. aureus*: 6.95	
101	**P**; n = 1, m = 14	2Br	MIC (µg/mL)	*E. coli*: >231.29; *E. faecalis*: >231.29; *P. aeruginosa*: >231.29; *S. aureus* (ATCC33592): >231.29; *S. aureus* (USA300-0114): 231.29; *S. aureus*: >231.29	
102	**R**; n = 1; m = 8	3Br	MIC (µg/mL)	*E. coli*: >245.04; *E. faecalis*: 245.04; *P. aeruginosa*: >245.04; *S. aureus* (ATCC33592): 245.04; *S. aureus* (USA300-0114): 122.52; *S. aureus*: 245.04	
103	**R**; n = 1; m = 9	3Br	MIC (µg/mL)	*E. coli*: 127.77; *E. faecalis*: 32.71; *P. aeruginosa*: 127.77; *S. aureus* (ATCC33592): 32.71; *S. aureus* (USA300-0114): 16.35; *S. aureus*: 32.71	
104	**R**; n = 1; m = 10	3Br	MIC (µg/mL)	*E. coli*: 17.03; *E. faecalis*: 17.03; *P. aeruginosa*: 34.05; *S. aureus* (ATCC33592): 8.51; *S. aureus* (USA300-0114): 8.51; *S. aureus*: 8.51	
105	**R**; n = 1; m = 11	3Br	MIC (µg/mL)	*E. coli*: 8.85; *E. faecalis*: 17.7; *P. aeruginosa*: 35.4; *S. aureus* (ATCC33592): 8.85; *S. aureus* (USA300-0114): 8.85; *S. aureus*: 4.42	
106	**R**; n = 1; m = 12	3Br	MIC (µg/mL)	*E. coli*: 9.19; *E. faecalis*: 9.19; *P. aeruginosa*: 18.37; *S. aureus* (ATCC33592): 4.59; *S. aureus* (USA300-0114): 4.59; *S. aureus*: 4.59	
107	**R**; n = 1; m = 14	3Br	MIC (µg/mL)	*E. coli*: 9.86; *E. faecalis*: 9.86; *P. aeruginosa*: 19.71; *S. aureus* (ATCC33592): 9.86; *S. aureus* (USA300-0114): 9.86; *S. aureus*: 4.93	
108	**R**; n = 1; m = 16	3Br	MIC (µg/mL)	*E. coli*: 82.92; *E. faecalis*: 42.12; *P. aeruginosa*: 329.04; *S. aureus* (ATCC33592): 42.12; *S. aureus* (USA300-0114): 21.06; *S. aureus*: 82.92	
109	**S**; n = 1, m = 5	3Br	MIC (µg/mL)	*E. coli*: 266.82; *E. faecalis*: 266.82; *P. aeruginosa*: >266.82; *S. aureus* (ATCC33592): 266.82; *S. aureus* (USA300-0114): 67.24; *S. aureus*: 67.24	
110	**S**; n = 1, m = 6	3Br	MIC (µg/mL)	*E. coli*: 69.89; *E. faecalis*: 35.5; *P. aeruginosa*: >277.32; *S. aureus* (ATCC33592): 35.5; *S. aureus* (USA300-0114): 8.87; *S. aureus*: 17.75	
111	**S**; n = 1, m = 7	3Br	MIC (µg/mL)	*E. coli*: 9.21; *E. faecalis*: 9.21; *P. aeruginosa*: 72.53; *S. aureus* (ATCC33592): 4.61; *S. aureus* (USA300-0114): 4.61; *S. aureus*: 2.3	
112	**S**; n = 1, m = 8	3Br	MIC (µg/mL)	*E. coli*: 4.77; *E. faecalis*: 9.55; *P. aeruginosa*: 19.09; *S. aureus* (ATCC33592): 4.77; *S. aureus* (USA300-0114): 4.77; *S. aureus*: 2.39	
113	**S**; n = 1, m = 9	3Br	MIC (µg/mL)	*E. coli*: 9.88; *E. faecalis*: 9.88; *P. aeruginosa*: 19.76; *S. aureus* (ATCC33592): 4.94; *S. aureus* (USA300-0114): 4.94; *S. aureus*: 2.47	
114	**S**; n = 1, m = 10	3Br	MIC (µg/mL)	*E. coli*: 10.55; *E. faecalis*: 10.55; *P. aeruginosa*: >329.82; *S. aureus* (ATCC33592): 10.55; *S. aureus* (USA300-0114): 10.55; *S. aureus*: 5.28	
115	**S**; n = 1, m = 10	3Br	MIC (µg/mL)	*E. coli*: 44.91; *E. faecalis*: 22.45; *P. aeruginosa*: 350.82; *S. aureus* (ATCC33592): 44.91; *S. aureus* (USA300-0114): 22.45; *S. aureus*: 175.41	
116	**T**; R_1_ = CH_3_, R_2_ = C_6_H_13_, R_3_ = CH_3_, R_4_ = CH_3_, n = 10	Cl	MIC (µg/mL)	*B. subtilis* (MTCC121): 7.8; *C. aaseri* (MTCC1962): 31.2; *C. albicans* (MTCC1637): 31.2; *C. albicans* (MTCC183): 15.6; *C. albicans* (MTCC277): 31.2; *C. albicans* (MTCC3017): 15.6; *C. albicans* (MTCC3018): 15.6; *C. albicans* (MTCC3958): 31.2; *C. albicans* (MTCC4748): 15.6; *C. albicans* (MTCC7315): 15.6; *C. albicans* (MTCC854): 15.6; *C. glabrata* (MTCC3019): 15.6; *C. krusei* (MTCC3020): 15.6; *C. parapsilosis* (MTCC1744): 15.6; *E. coli* (MTCC739): 31.2; *Issatchenikiahanoiensis* (MTCC4755): 15.6; *K. planticola* (MTCC530): 15.6; *M. luteus* (MTCC2470): 3.9; *P. aeruginosa* (MTCC2453): >125; *S. aureus* (MTCC96): 3.9; *S. aureus* (mLS16MTCC2940): 7.8	[112]
117	**T**; R_1_ = CH_3_, R_2_ = C_12_H_25_, R_3_ = CH_3_, R_4_ = CH_3_, n = 10	Cl	MIC (µg/mL)	*B. subtilis* (MTCC121): 15.6; *C. aaseri* (MTCC1962): 31.2; *C. albicans* (MTCC1637): 62.5; *C. albicans* (MTCC183): 31.2; *C. albicans* (MTCC277): 31.2; *C. albicans* (MTCC3017): 31.2; *C. albicans* (MTCC3018): 62.5; *C. albicans* (MTCC3958): 62.5; *C. albicans* (MTCC4748): 15.6; *C. albicans* (MTCC7315): 31.2; *C. albicans* (MTCC854): 31.2; *C. glabrata* (MTCC3019): 31.2; *C. krusei* (MTCC3020): 62.5; *C. parapsilosis* (MTCC1744): 62.5; *E. coli* (MTCC739): 15.6; *Issatchenikiahanoiensis* (MTCC4755): 31.2; *K. planticola* (MTCC530): 7.8; *M. luteus* (MTCC2470): 7.8; *P. aeruginosa* (MTCC2453): >125; *S. aureus* (MTCC96): 15.6; *S. aureus* (mLS16MTCC2940): 7.8	
118	**T**; R_1_ = CH_3_, R_2_ = C_18_H_37_, R_3_ = CH_3_, R_4_ = CH_3_, n = 10	Cl	MIC (µg/mL)	*B. subtilis* (MTCC121): >125; *C. albicans* (MTCC3017): >125; *E. coli* (MTCC739): >125; *K. planticola* (MTCC530): >125; *M. luteus* (MTCC2470): >125; *P. aeruginosa* (MTCC2453): >125; *S. aureus* (MTCC96): >125; *S. aureus* (mLS16MTCC2940): >125	
119	**T**; R_1_ = C_10_H_20_COOCH_3_, R_2_ = C_6_H_13_, R_3_ = CH_3_, R_4_ = CH_3_, n = 10	Cl	MIC (µg/mL)	*B. subtilis* (MTCC121): 7.8; *C. aaseri* (MTCC1962): 7.8; *C. albicans* (MTCC1637): 15.6; *C. albicans* (MTCC183): 15.6; *C. albicans* (MTCC277): 15.6; *C. albicans* (MTCC3017): 7.8; *C. albicans* (MTCC3018): 15.6; *C. albicans* (MTCC3958): 15.6; *C. albicans* (MTCC4748): 7.8; *C. albicans* (MTCC7315): 15.6; *C. albicans* (MTCC854): 31.2; *C. glabrata* (MTCC3019): 15.6; *C. krusei* (MTCC3020): 31.2; *C. parapsilosis* (MTCC1744): 15.6; *E. coli* (MTCC739): >125; *Issatchenikiahanoiensis* (MTCC4755): 15.6; *K. planticola* (MTCC530): >125; *M. luteus* (MTCC2470): 1.9; *P. aeruginosa* (MTCC2453): >125; *S. aureus* (MTCC96): 3.9; *S. aureus* (mLS16MTCC2940): 3.9	
120	**T**; R_1_ = C_10_H_20_COOCH_3_, R_2_ = C_12_H_25_, R_3_ = CH_3_, R_4_ = CH_3_, n = 10	Cl	MIC (µg/mL)	*B. subtilis* (MTCC121): >125; *C. albicans* (MTCC3017): >125; *E. coli* (MTCC739): >125; *K. planticola* (MTCC530): >125; *M. luteus* (MTCC2470): >125; *P. aeruginosa* (MTCC2453): >125; *S. aureus* (MTCC96): >125; *S. aureus* (mLS16MTCC2940): >125	
121	**T**; R_1_ = C_10_H_20_COOCH_3_, R_2_ = C_18_H_37_, R_3_ = CH_3_, R_4_ = CH_3_, n = 10	Cl	MIC (µg/mL)	*B. subtilis* (MTCC121): >125; *C. albicans* (MTCC3017): >125; *E. coli* (MTCC739): >125; *K. planticola* (MTCC530): >125; *M. luteus* (MTCC2470): >125; *P. aeruginosa* (MTCC2453): >125; *S. aureus* (MTCC96): >125; *S. aureus* (mLS16MTCC2940): >125	
122	**T**; R_1_ = C_6_H_11_, R_2_ = C_6_H_11_, R_3_ = CH_3_, R_4_ = CH_3_, n = 10	Cl	MIC (µg/mL)	*B. subtilis* (MTCC121): 7.8; *C. aaseri* (MTCC1962): 15.6; *C. albicans* (MTCC1637): 31.2; *C. albicans* (MTCC183): 15.6; *C. albicans* (MTCC277): 15.6; *C. albicans* (MTCC3017): 15.6; *C. albicans* (MTCC3018): 15.6; *C. albicans* (MTCC3958): 15.6; *C. albicans* (MTCC4748): 15.6; *C. albicans* (MTCC7315): 15.6; *C. albicans* (MTCC854): 31.2; *C. glabrata* (MTCC3019): 31.2; *C. krusei* (MTCC3020): 31.2; *C. parapsilosis* (MTCC1744): 15.6; *E. coli* (MTCC739): >125; *Issatchenikiahanoiensis* (MTCC4755): 15.6; *K. planticola* (MTCC530): >125; *M. luteus* (MTCC2470): 7.8; *P. aeruginosa* (MTCC2453): >125; S. aureus(MTCC96): 3.9; *S. aureus* (mLS16MTCC2940): 3.9	
123	**T**; R_1_ = C_8_H_17_, R_2_ = C_8_H_17_, R_3_ = CH_3_, R_4_ = CH_3_, n = 10	Cl	MIC (µg/mL)	*B. subtilis* (MTCC121): 7.8; *C. aaseri* (MTCC1962): 15.6; *C. albicans* (MTCC1637): 31.2; *C. albicans* (MTCC183): 31.2; *C. albicans* (MTCC277): 15.6; *C. albicans* (MTCC3017): 15.6; *C. albicans* (MTCC3018): 31.2; *C. albicans* (MTCC3958): 31.2; *C. albicans* (MTCC4748): 15.6; *C. albicans* (MTCC7315): 15.6; *C. albicans* (MTCC854): 15.6; *C. glabrata* (MTCC3019): 31.2; *C. krusei* (MTCC3020): 31.2; *C. parapsilosis* (MTCC1744): 62.5; *E. coli* (MTCC739): >125; *Issatchenikiahanoiensis* (MTCC4755): 31.2; *K. planticola* (MTCC530): 31.2; *M. luteus* (MTCC2470): 3.9; *P. aeruginosa* (MTCC2453): >125; *S. aureus* (MTCC96): 3.9; *S. aureus* (mLS16MTCC2940): 7.8	

^1^ IZD: inhibition zone diameter; ^2^ MIC: minimum inhibitory concentration; ^3^ MBC: minimum bactericidal concentration; ^4^ EC_50_: Half maximal effective concentration; ^5^ MCPP: (R)-2-(4-chloro-o-tolyloxy)propionate; ^6^ 4-CPA: 4-chlorophenoxyacetate; ^7^ 2,4-D: 2,4-Dichlorophenoxyacetate, ^8^ MCPA: 2-methyl-4-chlorophenoxyacetate; ^9^ Clopyralid: 3,6-dichloro-2-pyridinecarboxylate; ^10^ Doc: docusate, bis(2-ethylhexyl) sulfosuccinate; ^11^ Sac: sacharinate.

**Table 9 ijms-25-05761-t009:** Overview of reported results of toxicity of esterquats.

No	Core; Cation	Anion	Parameter (Units)	Value	Ref.
1	**A**; R_1_ = tallow, R_2_ = CH_3_	Cl	LD_50_ rat (oral) (mg/kg)	>10,000	[139]
			LD_50_ rabbit (dermal) (mg/kg)	>2000	
			96-h LD_50_ zebrafish (mg/L)	5.2	
			24-h EC_50_ for *D. magna* (immobilization) (mg/L)	14.8	
			72-h EC_50_ for *R. Subcapitata* (growth) (mg/L)	1.2	
2	**A**; R_1_ = tallow, R_2_ = CH_3_	MCPA ^1^	LD_50_ rat (oral) (mg/kg)	2000	[12]
			96-h LD_50_ rainbow trout (mg/L)	7–17	
			48-h EC_50_ for *D. magna* (immobilization) (mg/L)	0.2–0.5	
			72-h EC_50_ for *R. Subcapitata* (growth) (mg/L)	1.6–1.9	
3	**A**; R_1_= tallow, R_2_ = C_2_H_5_OH	MeSO_3_	LD_50_ rat (oral) (mg/kg)	>4480	[140]
			LD_50_ rat (dermal) (mg/kg)	>2000	
			96-h LD_50_ rainbow trout	1.91	
			48-h EC_50_ for *D. magna* (immobilization) (mg/L)	2.23	
			72-h EC_50_ for *S. Subcapitatus* (growth) (mg/L)	2.14	
4	**B**; n = 1, R_1_ = C_2_H_5_, R_2_ = CH_3_	Dic ^2^	48-h EC_50_ for *D. magna* (immobilization) (mg/L)	395	[133]
			48-h EC_50_ for *A. fransiscana* (immobilization) (mg/L)	>1000	
			72-h EC_50_ for *C. Vulgaris* (growth) (mg/L)	>1000	
5	**B**; n = 1, R_1_ = C_10_H_21_, R_2_ = CH_3_	Dic	48-h EC_50_ for *D. magna* (immobilization) (mg/L)	7.11	
			48-h EC_50_ for *A. fransiscana* (immobilization) (mg/L)	12.1	
			72-h EC_50_ for *C. Vulgaris* (growth) (mg/L)	88.9	
6	**B**; n = 1, R_1_ = C_16_H_33_, R_2_ = CH_3_	Dic	48-h EC_50_ for *D. magna* (immobilization) (mg/L)	0.4	
			48-h EC_50_ for *A. fransiscana* (immobilization) (mg/L)	263	
			72-h EC_50_ for *C. Vulgaris* (growth) (mg/L)	0.96	
7	**B**; n = 1, R_1_ = C_12_H_25_, R_2_ = CH_3_	SCN	72-h EC_50_ for *R. Subcapitata* (growth) (mg/L)	0.27	[21]
8	**B**; n = 1, R_1_ = C_12_H_25_, R_2_ = CH_3_	Doc ^3^		0.58	
9	**B**; n = 1, R_1_ = C_14_H_29_, R_2_ = CH_3_	SCN		0.54	
10	**B**; n = 1, R_1_ = C_14_H_29_, R_2_ = CH_3_	Doc		0.74	
11	**B**; n = 1, R_1_ = C_2_H_5_, R_2_ = C_4_H_9_	Sac ^4^		4.71	
12	**B**; n = 3, R_1_ = C_2_H_5_, R_2_ = C_4_H_9_	Sac		12.83	
13	**C**; m = 7	2Cl	48-h EC_50_ for *D. magna* (immobilization) (mg/L)	50	[111]
14	**C**; m = 9	2Cl		7.5	
15	**D**; m = 7	3Cl		37	
16	**D**; m = 9	3Cl		8.1	
17	**D**; m = 11	3Cl		1.5	

^1^ MCPA: 2-methyl-4-chlorophenoxyacetate; ^2^ Dic: dicamba, 3,6-dichloro-2-methoxybenzoate; ^3^ Doc: docusate, bis(2-ethylhexyl) sulfosuccinate; ^4^ Sac: sacharinate.

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
