# Peer review of "Rethinking the Esterquats: Synthesis, Stability, Ecotoxicity and Applications of Esterquats Incorporating Analogs of Betaine or Choline as the Cation in Their Structure"

_ijms, 2024, doi:10.3390/ijms25115761_

Round 1

Reviewer 1 Report

Comments and Suggestions for Authors

Thank you for giving me the opportunity to review one of the manuscripts submitted to your journal. I read the manuscript carefully, and it took some time due to the wealth of content and data to review. After an evaluation, I believe that the article deserves to be accepted in its current form. I just have two comments for the authors.

First, I suggest that authors include a summary describing all the content of the review. This would provide readers with a clear overview of the structure of the document.

Second, I recommend that the authors include additional data regarding gemini ester quat. it is necessary to at least present the method of synthesis of these gemini, there will be many articles describing the method of synthesis as well as some biological properties.

Overall, I find the manuscript well-written and informative, and I think responding to these two comments would further strengthen its quality and impact.

Reviewer 2 Report

Comments and Suggestions for Authors

Dear Editor and Authors,

The review article manuscript entitled “Rethinking the Esterquats: Synthesis, Stability, Ecotoxicity, and Applications of Esterquats Incorporating Analogs of Betaine or Choline as the Cation in Their Structure” by Shamshina, J.L. et al. describes the synthesis and possible applications of esterquats – quaternary ammonium salts and ionic liquids possessing ester functionality. Esterquats are based on choline and betaine cation and their analogs derivatized by an ester group (in the case of choline by esterification of the free OH group and in the case of betaine by esterification of carboxylate), which in many cases facilitates their hydrolysis and biodegradation. They are important alternatives to analogous unfunctionalized tetraalkylammonium cations which find many important industrial applications but are of great concern due to their stability and therefore pollution and toxicity. These esterquats alone, to the best of my knowledge, have only been reviewed once (in an article that needs to be cited) by Liwarska-Bizukojc in 2009 (https://www.cabidigitallibrary.org/doi/full/10.5555/20093311897), thus there is a great need for another review article. The article is generally well-structured and written, particularly in respect to chemical transformation and synthesis, but some major revisions are necessary, particularly in respect to the justification of esterquats’ ecotoxicity, toxicity, and biological activity.

1st comment: The part considering ecotoxicity should be better elaborated, defined, maybe in a separate chapter as it is one of the topics of the review or together with toxicity studies (separate from biodegradation and antimicrobial activity). There is one other, even more important aspect to be addressed here – acetylcholine and other acylated choline cations are just mentioned as naturally occurring and biodegradable cations (lines 160-164). Choline is indeed an essential nutrient and non-toxic (although for humans there is also a daily uptake limit), but acylated choline (even acetylcholine) is not due to their physiological activity. The literature results studying the toxicity of these compounds as well as other esterquats particularly containing core structures of choline, triethanolamine, betaine, and other analogs should be better represented as they pretend to show advantages in respect to other quaternary ammonium analogs. For example, the choline-based esterquates (lines 167-170) are described as possessing: “The excellent ecotoxicological properties…” although in references cited (ref. 54-56) except antimicrobial and antioxidant activity (ref.56), only synthesis is given with no other studies. How are the ecotoxicological properties evaluated? The authors should cite available information in the review correctly or if they find unfounded claims based on insufficient data in the literature (particularly in somewhat outdated chemistry journals, like for example in many articles from the beginning of the century), they should express their phrases neutrally explaining that the authors of these papers claim the ecotoxicological properties although by more contemporary recommendations they may not be considered so. For the recommendation for ecotoxicity characterization of chemicals they can follow for example the recent article of Owsianiak, M. et al. Chemosphere 2023, 310, 136807 (https://doi.org/10.1016/j.chemosphere.2022.136807), or other recent literature that should also be cited in the article. In the case of the evaluation of toxicity of ionic liquids for example recently a very nice review has been published by Gonçalves A.R.P et al. Int. J. Mol. Sci. 2021, 22, 5612. (https://doi.org/10.3390/ijms22115612).

2nd comment: The chapter about herbicidal formulation (2.1.1) seems to suffer from the same problem regarding toxicity. To my knowledge, some herbicides like dicamba, 2,4D, and others have a dubious toxicity profile themselves. What can be mentioned about their herbicidal formulation in this respect? Diverse chemical structures and possibilities are very nicely described in this chapter, but what can be said about the toxicity profile of these novel compounds? A great number of articles about this work are cited but it seems that there are no toxicity studies or profiling. If this information exists it should be included in this review. If it does not exist, this fact should be mentioned so that in the future the authors or investigators working in this field should give more attention to this respect.

3rd comment: Similar recommendations can be applied in chapter 2.2.2 about other biologically active systems. This information is probably more available in the case of compounds used as drug delivery systems and should be presented.

4th comment: In Chapter 2.3. (Synthesis of betaine-type esterquats), it is not clear the correlation of betaine-type esterquats and DES. What did the authors pretend with this comparison? Betaine is indeed mentioned in literature as a powerful HBA in DES, but this is true for carboxylate-free betaine (not an ester). The same is true for choline-based DES and choline-type esterquats where the anion contribution as HBA is crucial. If the esterquats form DES, it is certainly due mostly to the HBA activity of an anion not esterquat cation alone. Please clarify.

5th comment: Biological test chapters: biodegradation of esterquats (chapter 4.1) and toxicity of esterquats (chapter 4.2) somewhat compensate for the lack of an ecotoxicity chapter but in my opinion, it is insufficient as biodegradability and ecotoxicity are not the same. The toxicity of esterquats chapter seems to refer exclusively to antibacterial properties of these compounds and thus the suggestion is to rename this chapter accordingly. Any other data related to ecotoxicity/toxicity demonstrated with eukaryotic cells (if any) or living organisms should be joined in a separate chapter.

6th comment: The author claims (lines 99-102): “The increased interest in esterquats from the largest chemical companies, such as BASF, Henkel, Procter & Gamble, or Evonik serves as an indicator of their great usability and sufficient safety, which would potentially translate to their widespread commercialization and scale-up production [37–41].” as well as (lines 727- 731)“Numerous scientific patents disclose the use of esterquats as components of creams and lotions, shampoos, conditioners or various other skin cosmetics. Beside improving their stability, esterquats have also been proposed as safer replacements for non-natural origin ingredients such as silicones. Due to rising awareness of the customers and their growing requirements, this action is part of the modern marketing campaign promoting eco-friendliness [38,39,134,135].” In accordance with the presented results (i.e. biodegradability), it is certainly true that some of esterquats are better alternatives to classical tetraalkylammonium salts, however, as mentioned above, the lack of tests, particularly more complete toxicity tests should not just encourage their widespread use just because “the largest chemical companies” use them. In my opinion, the objective of this review is to encourage continued caution and further evaluation in this respect.

7th comment: In chapter 3 (Susceptibility to hydrolysis of choline-type and betaine-type esterquats), alkanoyl choline and alkyl-betaine transformation under methanogenic conditions was discussed (figure 8). I believe the hydrolysis and degradation of esterquats under aerobic conditions should be discussed as well as it seems to be more relevant, particularly as their application is mostly seen in agriculture, as alternative nontoxic solvents etc. The degradation pathway of choline and betaine under aerobic conditions is also sometimes problematic (see also betaine removal, aerobic degradation of betaine and choline, particularly in the sea, formation of trimethylamine oxide (TMAO)…

Thank you for your attention.

Round 2

Reviewer 2 Report

Comments and Suggestions for Authors

Dear Editor and Authors,

The authors have favorably responded to most of my comments and suggestions and have modified the manuscript accordingly. I believe the manuscript, in its present form, can be accepted for publication.

Thank you for your attention